# An improved zebrafish transcriptome annotation for sensitive and comprehensive detection of cell type-specific genes

Nathan D Lawson[1]*, Rui Li[1], Masahiro Shin[1], Ann Grosse[1†], Onur Yukselen[2], Oliver A Stone[3], Alper Kucukural[4,5], Lihua Zhu[1,4,5]

[1]Department of Molecular, Cell and Cancer Biology, University of Massachusetts Medical School, Worcester, United States; [2]Bioinformatics Core, University of Massachusetts Medical School, Worcester, United States; [3]Department of Physiology, Anatomy and Genetics, University of Oxford, Oxford, United Kingdom; [4]Program in Molecular Medicine, University of Massachusetts Medical School, Worcester, United States; [5]Program in Bioinformatics and Integrative Biology, University of Massachusetts Medical School, Worcester, United States

**Abstract** The zebrafish is ideal for studying embryogenesis and is increasingly applied to model human disease. In these contexts, RNA-sequencing (RNA-seq) provides mechanistic insights by identifying transcriptome changes between experimental conditions. Application of RNA-seq relies on accurate transcript annotation for a genome of interest. Here, we find discrepancies in analysis from RNA-seq datasets quantified using Ensembl and RefSeq zebrafish annotations. These issues were due, in part, to variably annotated 3' untranslated regions and thousands of gene models missing from each annotation. Since these discrepancies could compromise downstream analyses and biological reproducibility, we built a more comprehensive zebrafish transcriptome annotation that addresses these deficiencies. Our annotation improves detection of cell type-specific genes in both bulk and single cell RNA-seq datasets, where it also improves resolution of cell clustering. Thus, we demonstrate that our new transcriptome annotation can outperform existing annotations, providing an important resource for zebrafish researchers.

**\*For correspondence:**
nathan.lawson@umassmed.edu

**Present address:** †Scholar Rock, Inc, Cambridge, United States

**Competing interests:** The authors declare that no competing interests exist.

## Introduction

The zebrafish has become an ideal animal model for studying processes related to developmental biology (e.g. *Wilkinson and van Eeden, 2014*). Due to its hardy nature and fecundity, it is highly amenable to a variety of manipulations, including both forward and reverse genetic approaches (*Lawson and Wolfe, 2011*). The power of these applications is enhanced by the unique characteristics of the zebrafish embryo. Most notably, zebrafish embryos are transparent and develop externally, allowing direct observation and facile technical manipulation (*Beis and Stainier, 2006*). Moreover, the initial steps of zebrafish development are very rapid, with gastrulation completed by 24 hr post-fertilization (hpf) and a complete circulatory system online just 12 hr later. By 5 days post-fertilization (dpf), most organ systems are fully functional. Together, these characteristics of zebrafish life history have contributed to its establishment as a leading model to study developmental biology, while expanding its utility to the modeling of human diseases (*Lam and Peterson, 2019*).

In most model systems, a common analytical approach is to profile changes in gene expression following a particular experimental manipulation, or in comparison between wild type and mutant siblings. With the initial advent of microarrays and, more recently, RNA-sequencing (RNA-seq), this

approach can be applied at a transcriptome-wide level, enabling researchers to gain crucial insights into molecular defects associated with a particular experimental condition (*Smith, 2003*; *Wang et al., 2009*). Such analysis can reveal important effectors for signaling pathways, downstream targets for transcription factors, or identify changes in cellular composition in a complex tissue. More recently, several groups have adapted RNA-seq to allow molecular profiling of transcriptomes from single cells (*Klein et al., 2015*; *Macosko et al., 2015*). Regarding zebrafish, this powerful method has been applied in a number of novel ways. Initial efforts integrated single-cell RNA-seq (scRNA-seq) with a publicly available expression pattern database to provide a predictive spatial gene map for early stage zebrafish embryos (*Satija et al., 2015*). Recent application of scRNA-seq at successive developmental stages has allowed unprecedented molecular insights into lineage relationships during embryogenesis (*Farrell et al., 2018*; *Wagner et al., 2018*). More advanced applications have combined scRNA-seq with cas9-induced 'scars' to allow molecular lineage tracing and define cellular hierarchies in the brain (*McKenna et al., 2016*; *Raj et al., 2018*). Together, these studies demonstrate the power of combining the unique benefits of zebrafish embryogenesis with scRNA-seq to gain new insights into vertebrate development.

The application of RNA-seq relies on the ability to accurately and reliably process large datasets consisting of millions of short fragment reads (*Wang et al., 2009*). This approach usually relies on the availability of a genomic sequence to which reads are initially mapped. Subsequently, the positions of mapped reads matching a corresponding transcriptome annotation are used for gene assignment and quantification across different experimental conditions. For most commonly used model systems, gene models and transcriptome annotations are now generally robust, allowing the universal application of RNA-seq-based analysis with a variety of standardized computational pipelines. However, in this study, we have found several deficiencies in current transcriptome annotations for the zebrafish genome that can negatively impact RNA-seq analyses. To address these issues, we have generated a new zebrafish transcriptome annotation, and demonstrated its improved performance over existing annotations in several contexts, including analysis of bulk and scRNA-seq datasets from multiple cell types. Together, our results demonstrate that this annotation will be a valuable resource for the zebrafish community.

## Results

### Discrepancies in RNA-seq analysis between zebrafish Ensembl and RefSeq annotations

We have previously used RNA-seq to profile endothelial cell gene expression in zebrafish embryos (*Quillien et al., 2017*; *Whitesell et al., 2019*). In unpublished observations from those studies, we noted discrepancies between identical datasets separately quantified using Ensembl (version 95, hereafter referred to as Ens95) and RefSeq (GCF_000002035.6_GRCz11) transcriptome annotations. For example, we performed RNA-seq on mCherry-positive ($kdrl^{pos}$) and -negative ($kdrl^{neg}$) cells isolated by fluorescence-activated cell sorting (FACS) from $Tg(kdrl:HRAS-mCherry)^{s896}$ embryos, which express membrane-localized red fluorescence protein in endothelial cells (*Chi et al., 2008*; *Whitesell et al., 2019*). Use of the RefSeq annotation to quantify gene expression led to the identification of 1780 genes enriched in $kdrl^{pos}$ compared to $kdrl^{neg}$ cells, while Ens95 identified 1632 genes ($log_2$ fold change >1, padj <0.05, n = 3; *Figure 1A,B*; *Figure 1—source data 1*; *Table 1*). The intersection of $kdrl^{pos}$-enriched genes from each dataset using NCBI ID as a common identifier yielded 132 and 222 genes that were identified only by Ens95 or RefSeq, respectively, while 1407 were commonly identified as such in both annotation (*Figure 1C*, *Figure 1—source data 2*). Notably, common IDs were lacking for a sizable proportion of genes in both datasets (for example, 4215 out of 25704 in Ens95 lacked a matching NCBI ID; *Table 1*, *Figure 1—source data 1*), suggesting that each annotation may be missing genes. Importantly, genes detected as enriched only by Ens95 or RefSeq included those with previously known functional relevance. Among these were the artery-specific gene, *ephrin-B2a* (*efnb2a*; *Lawson et al., 2001*), and the *SRY-box transcription factor 17* (*sox17*; *Chung et al., 2011*), which were only identified as differential by Ens95 (*Figure 1A,B*, *Figure 1—source data 2*). By contrast, *slc2a1b*, one of two duplicated zebrafish genes that encodes the GLUT1 glucose transporter, as well as other blood-brain barrier markers, such as *solute carrier family 7 member 5* (*slc7a5*; *Boado et al., 1999*), were significantly enriched in $kdrl^{pos}$ cells using RefSeq,

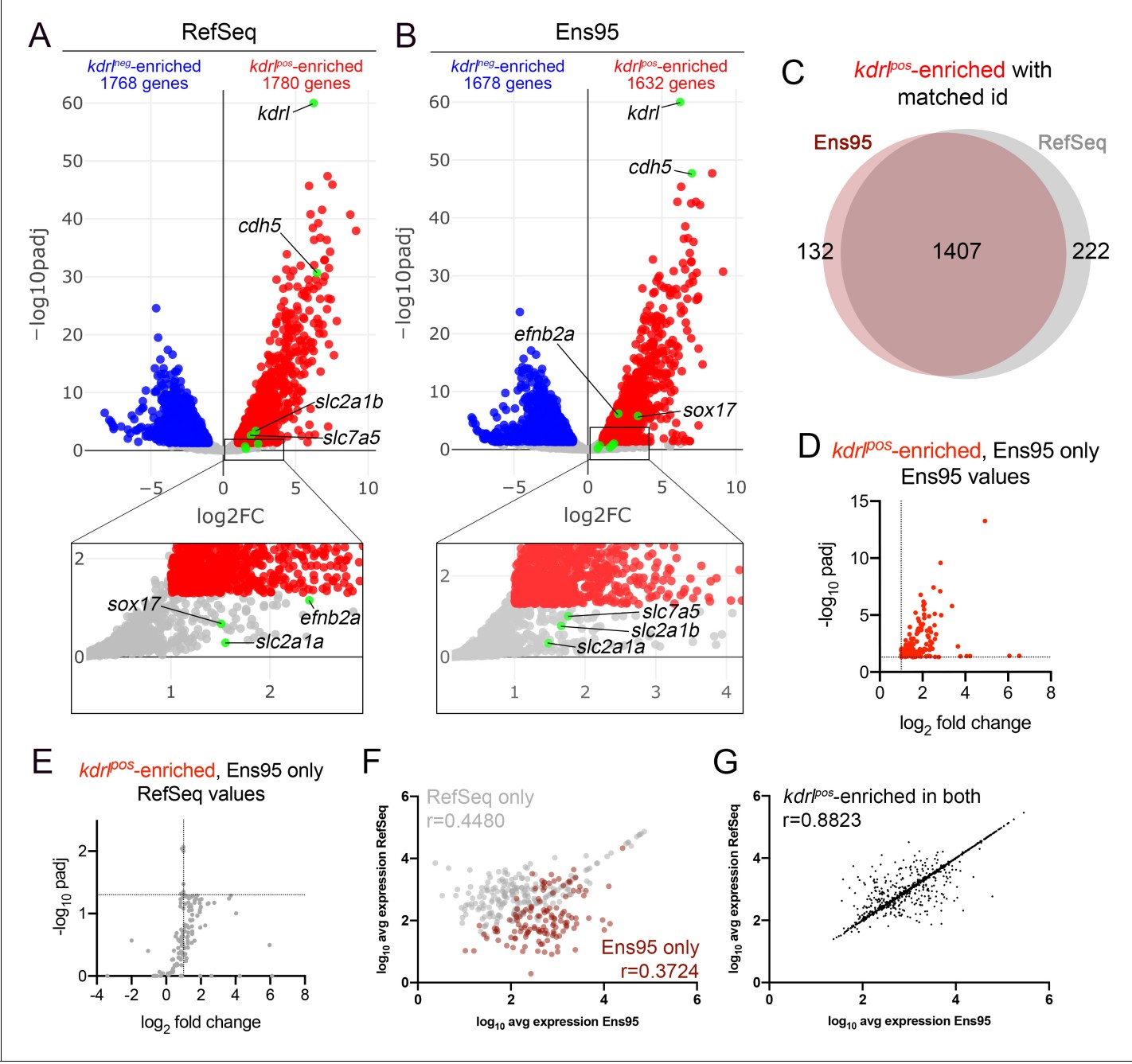

**Figure 1.** Comparison of Ensembl and RefSeq zebrafish transcriptome annotations for bulk RNA-seq analysis. (A, B) Volcano plots showing differentially expressed genes from *Tg(kdrl:HRAS-mCherry)*[s896]-positive and negative (*kdrl*[pos] and *kdrl*[neg]) cells identified using RNA-seq reads quantified with (A) RefSeq or (B) Ensembl, version 95 (Ens95) transcript annotations. Genes with significant enrichment (padj <0.05) are shown as red or blue (log$_2$ fold change >1 or <-1, respectively) across replicate samples (n = 3). Grey dots are genes that fall below statistical cutoffs. Green dots indicate selected genes previously found to be enriched in endothelial cells in zebrafish or other models. (C) Venn diagram illustrating the intersection of genes with a common NCBI ID in Ens95 and RefSeq found significantly enriched in *kdrl*[pos] cells using either annotation. (D, E) Plots of commonly annotated genes identified as *kdrl*[pos]-enriched only by Ens95 with indicated values from (D) Ens95 or (E) RefSeq. (F, G) Correlation of expression levels from indicated annotation for *kdrl*[pos]-enriched genes identified (F) selectively as such by Ens95 (maroon) or RefSeq (grey) only, or (G) both annotations. Data are not normally distributed, Spearman correlation, r values are indicated.

The online version of this article includes the following source data and figure supplement(s) for figure 1:

**Source data 1.** DESeq2 output for *kdrl*[pos] and *kdrl*[neg] RNA-seq quantified with RefSeq (GCF_000002035.6_GRCz11; worksheet 1) or Ensembl, v95 (worksheet 2).

**Source data 2.** Intersection of *kdrl*[pos]-enriched genes from RefSeq and Ens95 commonly annotated by NCBI ID.

*Figure 1 continued on next page*

*Figure 1 continued*

**Figure supplement 1.** Comparison of Ens95 and RefSeq zebrafish transcriptome annotations for bulk RNA-seq analysis.

**Figure supplement 1—source data 1.** DESeq2 output for *pdgfrb^pos* and *pdgfrb^neg* RNA-seq quantified with RefSeq (GCF_000002035.6_GRCz11) or Ensembl, v95.

**Figure supplement 1—source data 2.** Intersection of *pdgfrb^pos*-enriched genes from RefSeq and Ens95 commonly annotated by NCBI ID.

but not Ens95 (*Figure 1A,B*, *Figure 1—source data 2*). Interestingly, *slc2a1a*, a second duplicate GLUT1 ortholog (previously referred to as '*glut1b*'; *Tseng et al., 2009*; *Umans et al., 2017*) was not identified as differential by either annotation, despite previous reports of its endothelial expression (*Figure 1A,B*, *Figure 1—source data 1*; *Umans et al., 2017*). By contrast, well-known endothelial markers, such as *cadherin5* (*cdh5*) and *kdrl* itself, were robustly detected as enriched by both annotations (*Figure 1A,B*).

Subtle differences in Ens95 and RefSeq transcript annotations might affect data normalization and differential analysis, slightly shifting adjusted p-values and causing genes to fall just below statistical cutoffs. However, this was not the case for many of the genes identified selectively by either annotation. For example, 58 out of 132 commonly annotated *kdrl^pos* genes enriched only in Ens95 showed an adjusted p-value over 5-fold greater than the standard cutoff (0.05) when quantified using RefSeq (compare *Figure 1D,E*, *Figure 1—source data 2*). Moreover, expression levels from each annotation for genes selectively identified as *kdrl^pos*-enriched only using Ens95 or RefSeq showed a poor correlation, with significantly higher expression in the annotation in which the gene was identified as differential (*Figure 1F*, *Figure 1—figure supplement 1A*, *Figure 1—source data 2*). By contrast, levels of *kdrl^pos*-enriched genes for genes commonly identified as differentially expressed displayed a high degree of correlation between Ens95 and RefSeq quantifications (*Figure 1G*).

To rule out that these discrepancies were unique to endothelial genes, we analyzed RNA-seq data from *TgBAC(pdgfrb:citrine)^s1010* embryos. In this case, transgene-positive (*pdgfrb^pos*) cells include epidermis, cartilage, pericytes, and vascular smooth muscle cells (*Vanhollebeke et al., 2015*). As above, most *pdgfrb^pos*-enriched genes with a common NCBI identifier (1905) were differentially expressed compared to negative (*pdgfrb^neg*) cells using both annotations, while 172 and 248 genes were selectively identified as such by only Ens95 or RefSeq, respectively (*Figure 1—figure supplement 1B*, *Figure 1—figure supplement 1—source datas 1* and *2*; *Table 1*). Among the discrepant genes were well-known pericyte markers, such as *potassium inwardly-rectifying channel subfamily J member 8* (*kcnj8*) and *chondroitin sulfate proteoglycan 4* (*cspg4*), which encodes the proteoglycan epitope NG2 (*Bondjers et al., 2006*; *Ozerdem et al., 2001*). Both *kcnj8* and *cspg4* were *pdgfrb^pos*-enriched using RefSeq, but not Ens95 (*Figure 1—figure supplement 1C,D*, *Figure 1—figure supplement 1—source data 2*). By contrast, *angiopoietin 1* (*angpt1*) was identified as *pdgfrb^pos*-enriched only using Ens95 (*Figure 1—figure supplement 1C,D*, *Figure 1—figure supplement 1—source data 2*). The pericyte marker *ATP-binding cassette, sub-family C (CFTR/MRP), member 9* (*abcc9*), and *pdgfrb* itself, are found as *pdgfrb^pos*-enriched with both annotations (*Figure 1—figure supplement 1C,D*, *Figure 1—figure supplement 1—source data 2*). We noted

**Table 1.** Numbers of detected and differential genes from RNA-seq datasets analyzed with Ens95 and RefSeq annotations with matched identifiers.

| | Ens95 | | RefSeq | |
|---|---|---|---|---|
| | All | w/NCBI ID | All | w/Ens ID |
| *kdrl^pos* – all detected | 25704 | 21489 | 27516 | 21903 |
| *kdrl^pos*-enriched | 1632 | 1538 | 1780 | 1651 |
| *pdgfrb^pos* – all detected | 25699 | 21480 | 27598 | 21903 |
| *pdgfrb^pos*-enriched | 2186 | 2091 | 2323 | 2188 |
| Nr2f2^pos – all detected | 20516 | 17867 | 21788 | 18164 |
| Nr2f2^pos-enriched | 568 | 516 | 580 | 508 |

similar discrepancies for known smooth muscle genes, including the transcription factor *myocardin* (*myocd*), for which an annotation appears to be missing from Ens95 (data not shown), and *calponin 1, basic, smooth muscle, b* (*cnn1b*), both of which were selectively identified using RefSeq. Similar to *kdrl^pos*-enriched genes, 80 out of 172 of *pdgfrb^pos*-enriched genes identified using Ens95 showed padj values for matched RefSeq genes at least 5-fold over the statistical cutoff (*Figure 1—figure supplement 1E*, *Figure 1—figure supplement 1—source data 2*). Likewise, expression levels for genes selectively enriched using Ens95 or RefSeq showed poor correlation between annotations and higher expression levels in the respective annotation in which it was detected as differentially expressed (*Figure 1—figure supplement 1F,G*, *Figure 1—figure supplement 1—source data 2*). By contrast, the expression of *pdgfrb^pos*-enriched genes identified by both annotations displayed a high degree of correlation (*Figure 1—figure supplement 1F*). Taken together, these observations suggest underlying differences in zebrafish Ensembl and RefSeq annotations that contribute to inconsistencies in differential gene analysis from RNA-seq data.

## Incomplete 3′ UTRs models influence RNA-seq quantification using existing annotations

Since we constructed the RNA-seq libraries in the experiments noted above using oligo-dT priming (*Quillien et al., 2017*; *Whitesell et al., 2019*), they would yield reads with a bias to the 3′ end of transcripts. Therefore, we investigated whether discrepancies noted above were due to differences between 3′ untranslated region (UTR) annotations in Ens95 and RefSeq. For this purpose, we sought to assemble a reference set of transcripts with annotated 3′ UTRs from Ens95 and RefSeq. Strikingly, in the process of assembling this reference panel, we found that nearly 20% of all coding genes in Ens95 lacked any transcript with an annotated 3′ UTR, while a smaller proportion also lacked a stop codon (*Table 2*, *Figure 2—source data 1*). By contrast, only about 6% of RefSeq genes lacked a 3′ UTR, and 1% lack a stop codon (*Table 2*, *Figure 2—source data 1*). Importantly, missing 3′ UTRs appeared to contribute to the discrepancies observed in our RNA-seq analyses. For example, 64 of the 217 *kdrl^pos*-enriched genes identified only using RefSeq lacked an Ens95 3′ UTR, and these were detected at significantly lower levels in Ens95 compared to RefSeq (*Figure 2A*, *Figure 2—source data 1*). Similar results were found for *pdgfrb^pos*-enriched genes, where 94 out of 248 genes identified only using RefSeq lacked a 3′ UTR annotation in Ens95 (*Figure 2B*, *Figure 2—source data 1*). Among the discrepant *kdrl^pos*-enriched genes was the previously noted blood brain barrier (BBB) marker *slc7a5*, where lack of a 3′ UTR in Ens95 resulted in an approximately ten-fold difference in detected expression between the two annotations (*Figure 2C*; N.B.: unless otherwise indicated UCSC browser images show a single representative transcript bearing the longest 3′ UTR for a given gene; *Figure 1—source data 2*). Interestingly, the location of mapped reads from a publicly available RNA-seq dataset (referred to as GSE32900), as well the genomic position of *kdrl^pos* reads, suggested an even longer *slc7a5* 3′ UTR than that annotated in RefSeq (*Figure 2C*). A putative longer 3′ UTR would also be concordant with a '3P-seq' feature on the same strand, which we annotated using a public dataset that defines genomic locations for 3′ transcript ends based on a biochemically stringent technique (*Jan et al., 2011*; *Ulitsky et al., 2012*). Thus, some of the discrepancies noted above are likely due to a lack of 3′ UTR annotation for a considerable proportion of Ensembl genes.

We next compared 3′ UTR lengths in cases where there existed corresponding annotations for this feature in both Ens95 and RefSeq. For this purpose, we generated a reference set of transcripts for 16,558 genes (see Materials and methods for details; *Figure 2—source data 2*). In our reference panel, just over one-half of genes exhibited the same length 3′ UTR in both annotations (9077 out of

**Table 2.** Number of coding genes and proportion without stop codon or 3′ UTR in indicated annotation.

|  | Ens95 | RefSeq |
| --- | --- | --- |
| All Genes | 32520 | 30445 |
| annotated CDS | 25592 | 26120 |
| CDS, missing annotated stop codon | 1585 | 269 |
| CDS, missing annotated 3′ UTR | 4703 | 1580 |

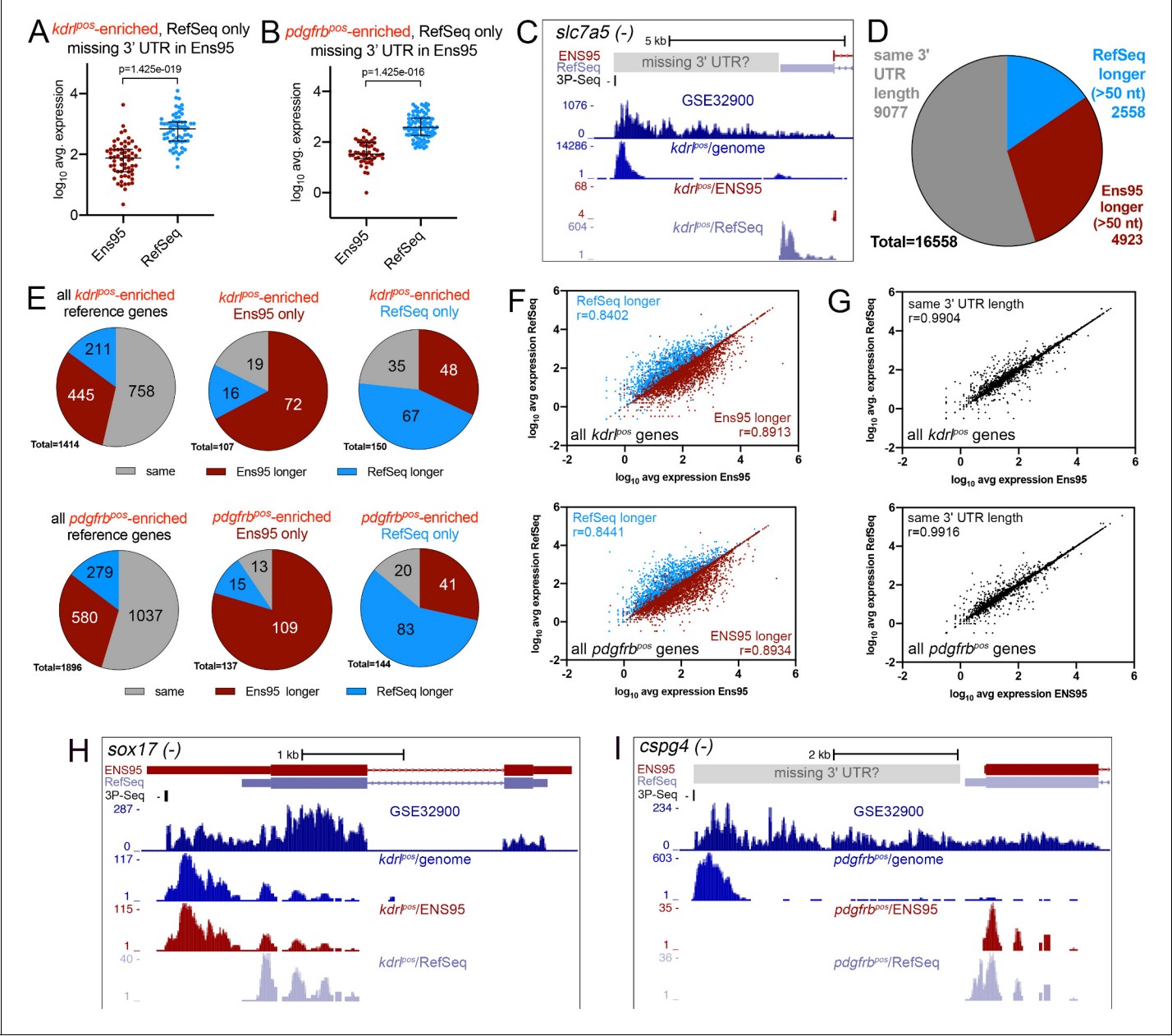

**Figure 2.** Incomplete 3' UTRs annotations contribute to discrepancies in RNA-seq analysis. (A, B) Log₁₀ average expression as quantified using indicated annotation for (A) *kdrl^pos*- or (B) *pdgfrb^pos*-enriched genes identified as such only in RefSeq and lacking an Ens95 3' UTR annotation. Expression levels for genes from each annotation with matched NCBI ID are shown in each case. Data are normally distributed (Shapiro-Wilks test), paired t-test, p values are indicated; n = 3 (i.e. each point represents an average value from three separate RNA-seq replicates). (C) UCSC browser image of *slc7a5* locus on the minus strand showing 3' UTR annotations from Ens95 and RefSeq. Mapped read depth from *kdrl^pos* cells on the genome, or assigned to each annotation are indicated, as is a 3P-seq feature. The GSE32900 track is consolidated RNA-seq reads from all stages indicated in *Figure 3A*. The location of a putative missing 3' UTR is indicated. (D) Pie chart showing numbers of reference genes with the same or longer 3' UTRs in each indicated annotation. (E) Pie charts showing the proportion of reference genes selectively identified as *kdrl^pos*- or *pdgfrb^pos*-enriched by Ens95 and RefSeq with indicated relative 3' UTR length. (F, G) Correlation plots showing log₁₀ average expression from *kdrl^pos* RNA-seq (n = 3) quantified with each annotation for matched reference genes with (F) longer Ens95 (maroon) or RefSeq (light blue) 3' UTR, or (G) same 3' UTR length. Data are not normally distributed, Spearman correlation, r values are indicated. (H, I) UCSC browser images of (H) *sox17* and (I) *cspg4* loci, both on the minus strand, showing 3' UTR annotations from Ens95 and RefSeq. Mapped read depth of RNA-seq from (H) *kdrl^pos* or (I) *pdgfrb^pos* cells captured for each annotation is indicated. Consolidated reads from GSE32900 and location of 3P-Seq features are indicated, as is putative missing 3' UTR in *cspg4*. The online version of this article includes the following source data and figure supplement(s) for figure 2:

**Source data 1.** Missing 3' UTR annotations in RefSeq and Ens95.

*Figure 2 continued on next page*

*Figure 2 continued*

**Source data 2.** Reference gene set for 3' UTR comparisons.
**Source data 3.** RNA-seq analysis of Nr2f2$^{pos}$ and NR2f2$^{neg}$ cells.
**Source data 4.** Transcript based-comparison of RefSeq and Ensembl annotations.
**Figure supplement 1.** Differences in 3' UTR lengths between Ens95 and RefSeq for discrepant *kdrl$^{pos}$*- and *pdgfrb$^{pos}$*-enriched genes.
**Figure supplement 2.** Analysis of RNA-seq reads from a random-primed library.

16558), while 4923 and 2558 genes showed longer 3' UTRs in Ens95 or RefSeq, respectively (*Figure 2D*, *Figure 2—source data 2*; 'longer' is defined as more than 50 nucleotides). Similar to genes with missing 3' UTRs, a large proportion of discrepant *kdrl$^{pos}$*- and *pdgfrb$^{pos}$*-enriched genes identified only in Ens95 or RefSeq also had a longer 3' UTR in the respective annotation (*Figure 2E*). Moreover, these 3' UTRs were, on average, five times longer when comparing annotations (*Figure 2—figure supplement 1A,B*, *Figure 2—source data 2*). Consistent with an effect on detection, levels of reference genes in *kdrl$^{pos}$* and *pdgfrb$^{pos}$* datasets quantified by Ens95 or RefSeq tended to be higher in the respective annotation predicting a longer 3' UTR and showed a lower correlation between annotations than those with the same 3' UTR length (*Figure 2F,G*, *Figure 2—source data 2*). Among these genes were several noted above as discrepant between Ens95 and RefSeq annotations when quantifying *kdrl$^{pos}$* and *pdgfrb$^{pos}$* datasets. For example, *sox17* exhibits a longer 3' UTR in Ens95, leading to improved detection at severalfold higher levels in *kdrl$^{pos}$* cells (*Figure 2H*, *Figure 1—source data 2*). We also noted additional cases where an accurate 3' UTR model is likely missing from both annotations. For example, the pericyte marker *cspg4* has a slightly longer 3' UTR in the RefSeq annotation (*Figure 2I*), where it is also detected as significantly enriched in *pdgfrb$^{pos}$* cells (*Figure 1—figure supplement 1C,D*). However, as noted above with *slc7a5*, the location of mapped reads from GSE32900 and our *pdgfrb$^{pos}$* data, as well as a 3P-Seq feature on the same strand, suggest a much longer 3' UTR (*Figure 2I*). Taken together, these results suggest that a large proportion of genes in current zebrafish transcriptome annotations do not have accurate 3' models, which can affect the interpretation of RNA-seq analysis.

Our discovery of 3' UTR discrepancies between Ens95 and RefSeq was likely facilitated by the 3' end-bias inherent in our RNA-Seq libraries. Therefore, we compared annotation performance using RNA-seq data from a random-primed library. In this case, we performed RNA-Seq on endothelial cells positive for the vein-specific transcription factor, Nr2f2 (nuclear receptor subfamily 2, group F, member 2), versus those positive for the *Tg(fli1a:egfp)$^{y1}$* transgene (see Materials and methods). Accordingly, we detected several known vein-specific genes enriched in Nr2f2$^{pos}$ cells including *fms related receptor tyrosine kinase 4* (*flt4*), *lymphatic vessel endothelial hyaluronic receptor 1b* (*lyve1b*), and *nr2f2* itself (*Figure 2—figure supplement 2A,B*, *Figure 2—source data 3*). Genes identified as Nr2f2$^{pos}$-enriched with matched NCBI IDs showed a similar degree of overlap from each annotation, as noted above for *kdrl$^{pos}$* and *pdgfrb$^{pos}$* analysis (*Figure 2—figure supplement 2C*). We also noted a poor correlation in detection levels for Nr2f2$^{pos}$-enriched genes selectively identified by either Ens95 or RefSeq compared to commonly identified genes (*Figure 2—figure supplement 2D*). In addition, we detected higher expression for discrepant genes in the annotation where the gene was significantly enriched, although the magnitude of difference in these cases was less than that of the 3' biased data (compare *Figure 2—figure supplement 2E* with *Figure 1—figure supplement 1A, G*). Accordingly, many of the discrepant Nr2f2$^{pos}$-enriched genes were associated with small shifts in adjusted p-values or log$_2$ fold change values. For example, only 10 out of 59 Nr2f2$^{pos}$-enriched genes identified by only Ens95 showed a greater than 5-fold padj compared to RefSeq values for the same genes (*Figure 2—figure supplement 2F*, *Figure 2—source data 3*). Furthermore, a comparison between discrepant reference Nr2f2$^{pos}$ genes did not show the same trend noted with *kdrl$^{pos}$* and *pdgfrb$^{pos}$* analysis regarding 3' UTR differences (*Figure 2—figure supplement 2G*, compare with *Figure 2E*). We noted a high degree of correlation between expression levels detected in each annotation regardless of differences in 3' UTR lengths, although we still noted a slight trend of longer 3'UTRs displaying slightly higher expression (*Figure 2—figure supplement 2H*, *Figure 2—source data 3*). These results suggest that differences in 3' UTR annotations between Ensembl and RefSeq may be somewhat mitigated when analyzing RNA-seq data from libraries constructed using random priming.

As noted above, nearly 20% of Ens95 genes lack a 3′ UTR annotation (see *Table 2*). For these Nr2f2[pos]-enriched genes, we did detect significant differences in expression levels between Ens95 and RefSeq annotations (*Figure 2—figure supplement 2I*). Notably, among these discrepant genes was the essential endothelial Ets transcription factor, *erg* (*Vijayaraj et al., 2012*), which lacks any UTR annotations in Ens95 compared to RefSeq, resulting in undersampling in Nr2f2[pos] cells when using Ens95 (*Figure 2—figure supplement 2J*). As a result, *erg* is identified as differentially enriched when using RefSeq, but not Ens95 for quantification (*Figure 2—source data 3*). Thus, the analysis of RNA-seq data obtained from random-primed libraries is less dependent on 3′ UTR annotation in existing Ensembl and RefSeq transcriptome annotations than when using 3′ biased data. However, relevant functional genes may still be missed when using Ensembl for quantification due to the complete absence of 3′ UTR annotations for many genes.

## Ensembl and RefSeq annotations are missing genes

During the analysis above, we noted that Ens95 and RefSeq zebrafish annotations lacked common identifiers for a sizable proportion of genes. Moreover, relevant lineage markers, such as the smooth muscle-specific gene *myocd*, were among genes missing from either annotation (see above and *Table 1*). To further investigate this issue, we performed a coordinate-based transcript-level comparison between Ens95 and RefSeq (see Materials and methods). This analysis revealed 3165 genes annotated by RefSeq that were missing from Ens95 (*Table 3*, *Figure 2—source data 4*). More than half of these missing genes encoded proteins and several hundred were curated in the Zebrafish Information Network (ZFIN; *Figure 2—source data 4*). Conversely, 2116 Ens95 genes were missing from RefSeq (*Table 3*, *Figure 2—source data 4*), including 670 protein-coding genes, 141 T-cell receptor genes, and 546 genes curated by ZFIN (*Table 3*, *Figure 2—source data 4*). These discrepancies may have been due to the older version of Ensembl that we used at the start of our studies. However, a comparison of Ens95 to a more current version (Ensembl, version 99; referred to as Ens99), revealed identical transcript models and no new genes that have been annotated in the intervening versions (*Figure 2—source data 4*), although some nomenclature has changed (data not shown). Thus, current zebrafish transcript annotations from RefSeq and Ensembl have considerable gaps that preclude a comprehensive assessment of gene expression and that may constrain follow-up studies when used for RNA-seq, or analyses focused on 3′ UTR sequences (e.g. miRNA target searches).

## Assembly of an improved zebrafish transcriptome annotation

Our analyses above demonstrated two major issues with existing zebrafish transcriptome annotations. First, 3′ UTR annotations are incomplete and differ between Ensembl and RefSeq. Second, each annotation is missing a sizable number of genes. To address these issues, we sought to construct a more complete zebrafish transcriptome annotation (*Figure 3A*). For this purpose, we used the GSE32900 RNA-seq datasets, which comprise libraries from zebrafish embryos at dome, bud, and shield stages, as well as 28 hr post-fertilization (hpf) and 2- and 5 days, post-fertilization (dpf). From these stages, we combined over 600 million paired-end, strand-specific reads (*Figure 3—source data 1*) and mapped them to GRCz11. We generated transcript and gene models using

**Table 3.** Coordinate-based transcriptome comparisons.

| Annotation | Ensembl 95 | RefSeq | V4.2 | V4.3 |
| --- | --- | --- | --- | --- |
| # genes | 32520 | 30445 | 39988 | 36351 |
| # transcripts | 59876 | 55182 | 115496 | 111842 |
| # exons | 335075 | 307538 | 414404 | 411330 |
| # RefSeq genes missing | 3165 | - | 173 | 7[c] |
| # Ensembl genes missing[a] | - | 2116 | 1133[b] | 957[d] |

a –RefSeq comparison with Ens95, V4 comparison with Ens99.

b – 956/1133 classified as rRNA, snRNA, snoRNA or sRNA.

c – left out from V4.2 add-back; see main text.

d – 956/957 are rRNA, snRNA, snoRNA, sRNA, or miscRNA; remaining protein coding gene is a sequence duplicate.

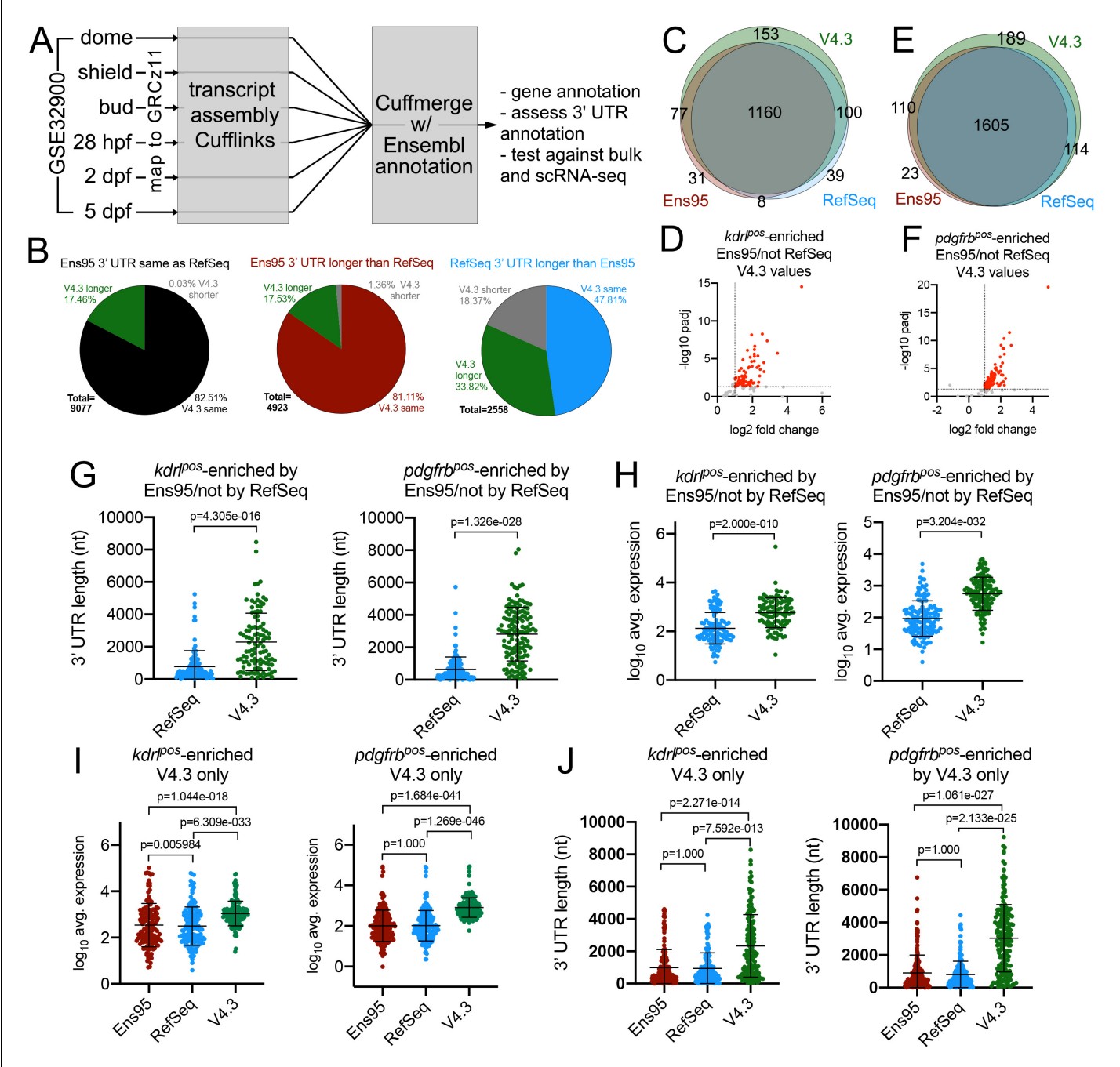

**Figure 3.** The V4.2 annotation improves detection of cell-type-specific genes from bulk RNA-seq data. (**A**) Schematic outline for generating a new zebrafish transcriptome annotation. See Results and Materials and methods sections for details. (**B**) Pie charts showing the proportion of reference genes with same, longer or shorter 3′ UTR in the V4.3 annotation compared to relative 3′ UTR length between Ens95 and RefSeq. (**C, E**) Venn diagrams showing intersection of reference genes with commonly annotated NCBI ID that are significantly enriched in (**C**) *kdrl*^pos^- or (**E**) *pdgfrb*^pos^-cells in each indicated annotation. (**D, F**) Volcano plots of reference genes with common NCBI ID identified as (**D**) *kdrl*^pos^- or (**F**) *pdgfrb*^pos^-enriched only by Ens95 in comparison to RefSeq. Indicated values are from the same genes quantified using V4.3. Red dots indicate $\log_2$ fold change >1 and adjp <0.05. (**G**) 3′ UTR lengths and (**H**) $\log_{10}$ average expression (n = 3) using RefSeq and V4.3 for reference genes in the indicated dataset. (**G, H**) Data are not normally distributed, Wilcoxon matched-pairs signed-rank test, p values are indicated. Error bars denote mean and standard deviation. (**I**) $\log_{10}$ average expression (n = 3) and (**J**) 3′ UTR lengths across all annotations for reference genes uniquely identified as enriched in indicated transgene-positive cell type using V4.3 ($\log_2$ fold change >1, padj <0.05). Data are not normally distributed. Friedman test to assess variance (p<0.0001 in all cases). Dunn's multiple comparison test was used for pairwise comparisons, p values are indicated. Error bars denote mean and standard deviation.

The online version of this article includes the following source data and figure supplement(s) for figure 3:

*Figure 3 continued on next page*

*Figure 3 continued*

**Source data 1.** List of SRA accession numbers, stages, and read numbers from GSE32900 for associated RNA-seq datasets used in this study.
**Source data 2.** List of manually-identified discrepancies in Ensembl gene annotation due to spurious fusionor overlapping transcripts.
**Source data 3.** RefSeq (worksheet 1) and Ens99 (worksheet 2) genes missing from the V4.2 annotation.
**Source data 4.** Novel genes from V4.2 genome annotation.
**Source data 5.** V4.3 gene information table, including unique LL ID numbers, associated Ens99 gene ID, NCBI ID, and ZFIN gene ID numbers, gene symbols, and gene names.
**Source data 6.** Output from DESeq2 analysis comparing $kdrl^{pos}$ and $kdrl^{neg}$ RNA-seq.
**Source data 7.** Output from DESeq2 analysis comparing $pdgfrb^{pos}$ and $pdgfrb^{neg}$ RNA-seq.
**Source data 8.** Worksheet 1 - Output from DESeq2 analysis comparing $Nr2f2^{pos}$ and $Nr2f2^{neg}$ RNA-seq.
**Figure supplement 1.** Ensembl naming conflicts and improved transcript diversity in V4.3.
**Figure supplement 2.** The V4.3 annotation improves the detection of cell-type-specific genes from bulk RNA-seq data.
**Figure supplement 3.** Analysis of $Nr2f2^{pos}$ and $Nr2f2^{neg}$ datasets using V4.3.

Cufflinks (*Trapnell et al., 2010*) and applied Cuffmerge (*Trapnell et al., 2010*) to combine them with Ens95 and RefSeq. With this approach, the naming conventions from the Ens95 annotation were applied to all gene models and remaining known genes named with RefSeq nomenclature. Initial transcriptome versions (versions 1 through 3; V1-V3) were generated to test and identify optimal parameters for transcript assembly (data not shown and see Materials and methods for details). During this process, we noted several nomenclature conflicts that arose due to spurious fusion transcripts. These were identified and removed through automated and manual annotation (see Materials and methods for details). These fusion transcripts led to misassignment of gene information in Ensembl and resulted in incorrect RNA-seq quantification, leading to shifts in detected gene expression (for examples, see *Figure 3—figure supplement 1*, *Figure 3—source data 2*). We also chose to remove ribosomal and small nuclear RNAs (but not miRNAs), which are present in Ensembl, but not in RefSeq.

The first working version of our annotation, referred to as version 4.2 (V4.2), comprised 115,496 transcripts from over 400,000 unique exons that give rise to nearly 40,000 genes (*Table 3*). A coordinate-based transcript comparison with RefSeq showed good coverage by V4.2 with only 173 RefSeq genes missing (*Table 3*, *Figure 3—source data 3*). Comparison with a more recent Ensembl transcriptome annotation (Ensembl 99, released January 2020) shows nearly 1200 genes missing from V4.2, although the majority of these (957/1133) are ribosomal or small nuclear RNAs, which we had removed from our annotations (*Table 3*, *Figure 3—source data 3*) and which are also absent from RefSeq. Over 6000 gene models in V4.2 did not overlap with existing annotations and are referred to using arbitrary 'XLOC' numbers assigned by Cufflinks (*Figure 3—source data 4*). Of these, approximately 1300 are transcripts that encode amino acid homology to zebrafish or human proteins, suggesting that these are previously unidentified paralogs or orthologs, although we cannot rule out the possibility that they may be pseudogenes. Of the XLOC genes, we also identified 45 that match previously annotated long non-coding RNAs not present in Ensembl or RefSeq (*Figure 3—source data 4*).

We subsequently added annotations for the missing Ens99 and RefSeq genes noted above from V4.2 (see *Table 3*; *Figure 3—source data 3*) to yield a V4.3 annotation. We also limited inclusion in V4.3 of XLOC genes to those that were supported by secondary analyses (see Materials and methods for details; *Figure 3—source data 4*). For end users concerned with potential mapping or quantification issues arising from novel gene annotations (e.g. due to the presence of pseudogenes, or repeat elements), XLOC genes can be easily removed from the annotation. Except for the cases noted above, V4.3 is missing only a single Ens99 gene (exclusive of small RNAs noted above) and 7 RefSeq genes (*Table 3*). The single Ens99 gene is an exactly duplicated sequence while missing RefSeq genes are those whose annotations are split between multiple chromosomes, with most cases having just a single or partial 5' exon as the second annotation (data not shown; *Figure 3—source data 3*). In these cases, we kept the larger annotation, which had been present in V4.2, that included the 3' UTR. The V4.3 annotation contains 111,842 transcripts from over 400,000 unique exons that comprise 36351 genes (*Table 3*). The transcript number in V4.3 is nearly double that in Ensembl or RefSeq and is accompanied by a considerable increase in the number of transcripts per gene (*Figure 3—figure supplement 1D*), suggesting increased transcript

isoform diversity. For example, Ensembl and RefSeq each predict a single transcript for the *erg* gene, each of which differs in intron-exon structure, as well as both 5' and 3' UTR (*Figure 3—figure supplement 1E*). The V4.3 annotation predicts four transcripts for *erg*, including the Ensembl and RefSeq transcripts, as well as new additional isoforms with alternative promoter or exon usage (*Figure 3—figure supplement 1E*).

To facilitate the use of V4.3 by researchers, we have made Gene Transfer Format (.gtf) files with exon, transcript, and gene-level coordinate annotations for both V4.2 and V4.3 available for download at zf-transcriptome.umassmed.edu. These files can be used for standard bulk or single-cell RNA-seq pipelines, with appropriate modification. We recommend that users apply the V4.3 annotation for their studies, given that it is more complete, although we include both for those interested in replicating studies described below. We have also compiled a gene information table containing matching Ensembl and RefSeq gene identifiers, along with ZFIN gene IDs, and chromosomal coordinates (*Figure 3—source data 5*). As described above, we noted some nomenclature conflicts between Ensembl and RefSeq, and there are likely to be more. Therefore, we also include comments in the information table regarding how Ensembl and RefSeq gene assignments were made. We recommend that users consider these when performing follow-up analyses to confirm that matching data imported from other sources is accurate. Since our annotation comprises genes missing from Ensembl or RefSeq, as well as potentially novel genes (e.g. those identified with 'XLOC' assignments), and would lack a corresponding identifier from those annotations, we also assigned a unique arbitrary gene identifier to every gene (referred to as an 'LL' ID number). The LL ID number is also assigned to entries annotated in the GTF allowing easy integration of information from this table (*Figure 3—source data 5*) with RNA-seq outputs following analysis. The gene information tables for both annotations are also available for download at zf-transcriptome.umassmed.edu.

## Improved 3' UTR coverage in the V4.3 transcriptome annotation

To quantify improvements in 3' UTR coverage in V4.3, we again relied on the reference gene set described above. For those reference genes where 3' UTR lengths were the same between Ens95 and RefSeq, we found that 17% (1584/9077) possessed a longer 3' UTR in the V4.3 annotation (*Figure 3B*, *Figure 2—source data 2*). Moreover, a notable proportion of reference genes with longer 3' UTRs in either Ens95 (688/4923) or RefSeq (865/2558), exhibited still longer 3' UTR annotations in V4.3 (*Figure 3B*, *Figure 2—source data 2*). In total, we identified 3129 genes out of the reference gene set (16558 total genes), where V4.3 predicted a longer 3' UTR than previously found in Ens95 or RefSeq for matching genes.

To determine how these extended annotations affect RNA-seq analysis, we quantified the 3' biased *kdrl:mcherry* and *pdgfrb:citrine* datasets using V4.3 and compared the output to that obtained from Ens95 and RefSeq. We observed a 20% to 30% increase in the number of $kdrl^{pos}$- or $pdgfrb^{pos}$-enriched genes over that identified when using Ens95 or RefSeq (*Figure 3—figure supplement 2A,B*; *Figure 3—source datas 6* and *7*; *Table 4*). The majority of these genes were those commonly annotated in both existing annotations, while a smaller proportion was present in only Ens95 or RefSeq. Only about 60 $kdrl^{pos}$- or $pdgfrb^{pos}$-enriched genes identified by V4.3 were those not previously identified in Ens95 or RefSeq annotations (i.e. XLOC-annotated genes; *Table 4*). By contrast, the number of $Nr2f2^{pos}$-enriched genes, identified using data from random-primed libraries, was similar when using V4.3 (*Figure 3—figure supplement 3A*, *Figure 3—source data 8*),

**Table 4.** Numbers of genes identified as enriched in indicated genotype with each annotation.

|  | $kdrl^{pos}$-enriched | $pdgfrb^{pos}$-enriched | $Nr2f2^{pos}$-enriched |
|---|---|---|---|
| Ens95 | 1632 | 2186 | 568 |
| RefSeq | 1780 | 2323 | 580 |
| V4.3 | 2141 | 2794 | 613 |
| annotated in Ens95 and RefSeq | 1938 | 2612 | 523 |
| not annotated in Ens95 | 144 | 131 | 67 |
| not annotated in RefSeq | 119 | 113 | 54 |
| only annotated in V4.3 | 60 | 62 | 31 |

suggesting the improvements noted above were due, in part, to the expanded 3′ UTR coverage in our annotation.

To broadly compare the enriched genes from each annotation, we again utilized our reference gene set, which also contains information on 3′ UTR lengths. As noted above, there were discrepancies between differentially called reference genes between Ens95 and RefSeq for the *kdrl:mcherry* and *pdgfrb:citrine* datasets (108 Ens95 only, 139 RefSeq only for *kdrl^pos^*-enriched; 123 Ens95 only and 114 RefSeq only for *pdgfrb^pos^*-enriched). Importantly, the majority of these discrepancies were eliminated by using V4.3. For example, 77 out of 108 genes identified as enriched only by Ens95 in *kdrl^pos^* cells were now called as significant using V4.3 (*Figure 3C,D*). Similar results were observed with *pdgfrb^pos^*-enriched genes, where 110 out of 139 genes detected in Ens95 only were also now detected as significant with V4.3 (*Figure 3E,F*). In both cases, these genes were detected at significantly higher levels using the V4.3 annotation compared to RefSeq and exhibited much longer 3′ UTR annotations (*Figure 3G,H*, *Figure 2—source data 2*). We also noted that approximately 10% of all differentially called *kdrl^pos^* (153) and *pdgfrb^pos^* (189) genes found in our reference gene set were uniquely identified using V4.3, but not Ens95 or RefSeq, despite their presence in all three annotations (*Figure 3C,E*; *Figure 3—source datas 6* and *7*). As above, these genes were detected at significantly higher levels and displayed much longer 3′ UTRs in the V4.3 annotation, compared to Ens95 or RefSeq, (*Figure 3I,J*; *Figure 2—source data 2*). Although we noted similar discrepancies between Ens95 and RefSeq annotations in the Nr2f2^pos^ datasets (*Figure 3—figure supplement 3B*, *Figure 3—source data 8*), many of these were not resolved using V4.3, with most discrepant genes found just below statistical thresholds used to call Nr2f2^pos^-enriched genes (for example, see *Figure 3—figure supplement 3C*, *Figure 3—source data 8*). However, we did note a number of cases where longer 3′ UTR annotations in V4.3 helped to identify genes not otherwise detected as Nr2f2^pos^-enriched by one of the other annotations (*Figure 3—figure supplement 3D,E*, *Figure 3—source data 8*).

Among the genes identified as significantly enriched by V4.3 were all of the individual examples noted above as discrepancies between Ens95 and RefSeq (*Figure 3—figure supplement 2A,B*). These included endothelial genes such as *slc7a5*, which possessed a longer 3′ UTR in RefSeq and was initially identified as *kdrl^pos^*-enriched only in that annotation. In V4.3, *slc7a5* has a 3′ UTR approximately 5 kb longer than Ens95 or RefSeq and is detected at much higher levels (*Figure 3—figure supplement 2C*). Consequently, it is detected as significantly enriched in *kdrl^pos^* cells when using V4.3 (*Figure 3—figure supplement 2A*). We also identified genes expected to be in endothelial cells but missed as significantly enriched by both Ens95 and RefSeq. This included *slc2a1a*, which has a 3′ UTR approximately 3 kb longer in V4.3 than Ens95 and RefSeq (*Figure 3—figure supplement 2D*). Accordingly, we detected *slc2a1a* at significantly higher levels in V4.3 than the other annotations and as significantly enriched in *kdrl^pos^* cells when using V4.3 (*Figure 3—figure supplement 2A,D*). We noted similar improvements with discrepant *pdgfrb^pos^* genes, such as *cspg4*, which has a longer 3′ UTR in V4.3 when compared to Ens95 and RefSeq, concomitant with much higher levels of expression detected by V4.3 (*Figure 3—figure supplement 2B,E*). Notably, in each of these cases, the longer 3′ UTRs identified by V4.3 were supported by the presence of a 3P-seq feature (*Figure 3—figure supplement 2C–E*). Taken together, these results suggest that analysis of bulk RNA-seq data using the V4.3 annotation can provide a more comprehensive and consistent output than Ensembl or RefSeq annotations.

## V4.2 improves analysis of single-cell RNA-seq data in zebrafish

Single-cell RNA-seq (scRNA-seq) is increasingly being applied to zebrafish to address relevant developmental questions. In most cases, scRNA-seq relies on 3′ end-sequencing to identify and count transcripts in each cell. Based on the increased 3′ UTR coverage in our V4 annotations, we reasoned that they could provide improvements for analyzing scRNA-seq data. To determine if this was the case, we used a publicly available 3′ scRNA-seq dataset obtained from zebrafish embryos at five dpf using the 10x Genomics platform (*Farnsworth et al., 2020*). In this case, we separately quantified mapped reads from this dataset to Ens95 and V4.2 annotations using the CellRanger pipeline followed by filtering, clustering, and differential expression analysis using Seurat (*Stuart et al., 2019*). For cell clustering, we utilized the first 75 principal components, which contributes to a comparable amount of the cumulative variation in both cases (see Materials and methods, *Source code 1*, *Figure 4—figure supplement 1*).

After initial mapping with Cell Ranger, we observed an increase in the number of genes and unique molecular identifiers (UMI) captured per cell, along with an increase in total genes detected when using V4.2 compared to Ens95 (*Figure 4—source data 1*). Most other standard metrics were comparable between the two annotations. We subsequently identified 54 cell clusters from a total of 9373 cells when using Ens95-quantified data (*Figure 4A*). With V4.2, both the number of cells (11630) and the number of clusters (62) was increased (*Figure 4B*). This suggested that the increased 3' UTR coverage in V4.2 allowed more reads to be captured and used for quantification, thereby increasing the number of gene and cell calls used for clustering. Accordingly, we observed notable increases in the number of cluster-specific genes per cluster and numbers of cells per cluster when comparing Ens95- and V4.2-quantified data (*Figure 4—figure supplement 1C,D*).

The increased number of cell clusters from the V4.2 annotation versus Ens95 suggested that some clusters may be split as a consequence of the increased numbers of genes and cells (see below). This makes a more granular cluster-to-cluster comparison challenging. Therefore, to further compare differences between scRNA-seq output using the two annotations, we identified cell clusters that could be unambiguously identified and matched between Ens95 and V4.2. We identified two such clusters for further analysis from browsing cluster-specific gene expression signatures from each dataset (*Figure 4—source datas 2* and *3*). The first was a cartilage cell cluster defined by expression of *mia* (*MIA SH3 domain-containing*) in both Ens95 and V4.2-annotated datasets (*Figure 4A–C*; cluster 18 in *Figure 4—source data 2*, cluster 22 in *Figure 4—source data 3*). In each annotation, *mia*-positive (*mia^pos*) cells expressed several other genes found in cartilage or corresponding pharyngeal arch progenitors, including *matrilin 1* (*matn1*), *collagen, type II, alpha 1a* (*col2a1a*), and *fibroblast growth factor binding protein 2b* (*fgfbp2*; *Figure 4—figure supplement 2A*, *Figure 4—source datas 2* and *3*; *Askary et al., 2017*; *Xu et al., 2018*). We could similarly define an epidermis cell population in both annotations by *actinodin1* (*and1*) expression (*Figure 4A, B,D*; identified as cluster 11 in *Figure 4—source data 2* and cluster 15 in *Figure 4—source data 3*), along with co-expression of *actinodin2* (*and2*), and *muscle segment homeobox 2b* (*msx2b*; *Figure 4—figure supplement 2B*; *Figure 4—source datas 2* and *3*; *Akimenko et al., 1995*; *Zhang et al., 2010*). The cartilage cluster comprised a similar number of cells regardless of annotation (197 with Ens95, 196 with V4.2), while V4.2 identified more cells in the epidermis cluster compared to Ensembl (268 for V4.2, 240 for Ens95). The intersection of cluster-specific genes for cartilage and epidermis from each annotation indicated that most genes identified by Ensembl were likewise detected with V4.2 (*Figure 5A*; 139 cartilage genes, 104 epidermis genes, *Figure 5—source data 1*). However, 25% more cartilage- and 15% more epidermis-specific genes were detected when using V4.2 (*Figure 5A*). Reference genes from both clusters identified only using V4.2 exhibited significantly longer 3' UTR length in that annotation compared to Ens95 (*Figure 5B*). By contrast, those few cluster-specific genes identified only using Ens95 largely fell just below statistical cutoffs or were cases of gene names assigned to more than one Ens95 locus that collapsed into a single gene in V4.2. For example, *carboxypeptidase N* (*cpn1*) is assigned to two closely adjacent genes in Ens95 and this locus is considered as a single *cpn1* gene in V4.2, where it is detected as cluster-specific by V4.2 (data not shown and *Figure 5—source data 1*).

To further characterize the cluster-specific genes uniquely identified by V4.2, we took advantage of the fact that the *TgBAC(pdgfrb:citrine)^s1010* transgene is expressed in both cartilage and epidermis (*Vanhollebeke et al., 2015*, data not shown). Therefore, corresponding cell type-specific genes identified by scRNA-seq should be represented and will likely be enriched in the corresponding *pdgfrb^pos* bulk RNA-seq analysis described above. Notably, the majority of cartilage (41 out of 57) and epidermis (20 out of 30) cluster-specific genes identified by V4.2 were also significantly enriched in *pdgfrb^pos* cells (*Figure 5C*; *Figure 5—source datas 1* and *2*). Thus, these are likely to be bona fide cartilage and epidermis genes that are uniquely detected only by V4.2 in scRNA-seq data, despite most being commonly annotated in Ens95 as well. Interestingly, many of these genes were also detected as *pdgfrb^pos*-enriched in bulk RNA-seq data quantified using Ens95 (*Figure 5—figure supplement 1A,B*, *Source datas 1* and *2*), despite the failure to identify them as such by scRNA-seq. This is likely due to the more restricted location of reads in a 3' scRNA-seq library (i.e. at the terminal 3' end of the transcript) versus bulk RNA-seq, where even in the case of 3' biased libraries reads can still be found distributed throughout the transcript.

Among the genes selectively identified by V4.2 were those encoding notable signaling molecules, such as *SRY-box transcription factor 9a* (*sox9a*) and the Wnt signaling inhibitor *frizzled class receptor*

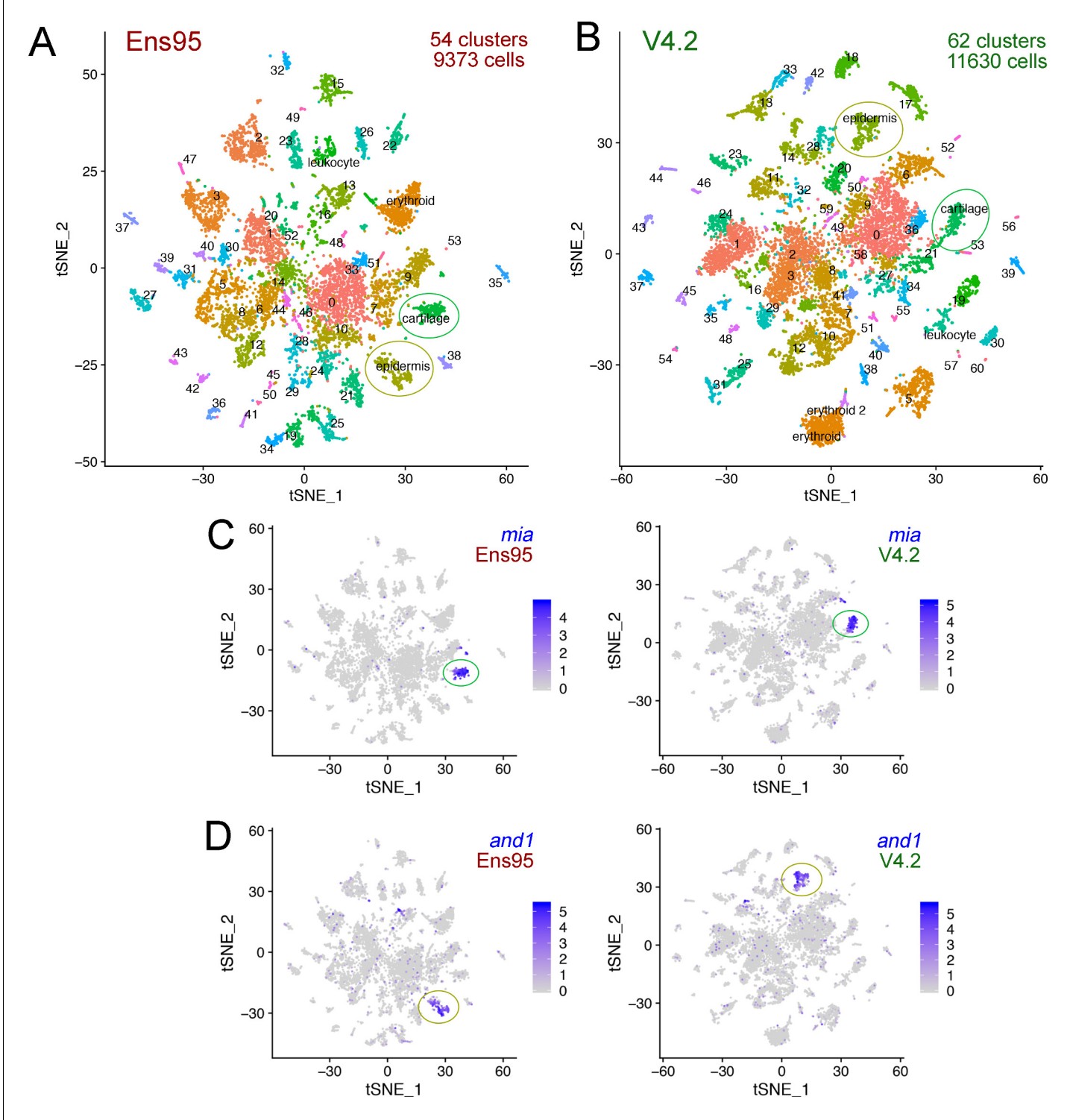

**Figure 4.** The V4.2 annotation increases the number of cells and clusters detected in single cellingle-cell RNA-seq analysis. (A, B) tSNE plots of cells from 5 day post fertilization (dpf) zebrafish embryos from the same mapped scRNA-seq reads quantified with (A) Ens95 or (B) V4.2. The total number of clusters and cells that passed filtering are shown. Clusters of interest noted in the text are named. Cartilage and epidermis clusters are circled. (C, D). tSNE plots showing restricted expression of (C) *mia* in cartilate cells and (D) *and1* in epidermis cells for both Ens95 and V4.2 annotations. Each cluster is indicated by a circle. Legends indicate log-transformed and normalized expression values.

The online version of this article includes the following source data and figure supplement(s) for figure 4:

**Source data 1.** Metrics from CellRanger output for data quantified using Ens95 and V4.2.

*Figure 4 continued on next page*

*Figure 4 continued*

**Source data 2.** Cluster-specific genes from whole embryo scRNA-seq at 5 days post fertilization (dpf) identified using Seurat from data quantified with Ens95.

**Source data 3.** Cluster-specific genes from whole embryo scRNA-seq at five dpf identified using Seurat from data quantified with V4.2.

**Figure supplement 1.** Standard variance and principal component comparisons for scRNA-seq analysis.

**Figure supplement 2.** Identification of cartilage and epidermis cell clusters in scRNA-seq data.

*9b* (*fzd9b*). Both of these genes have been previously found in cartilage cells or their progenitors (*Askary et al., 2017*; *Xu et al., 2018*) and were readily detected as significantly enriched in the cartilage cluster from V4.2-quantified data, but not Ens95 (*Figure 5D,E*, *Figure 5—source data 1*).

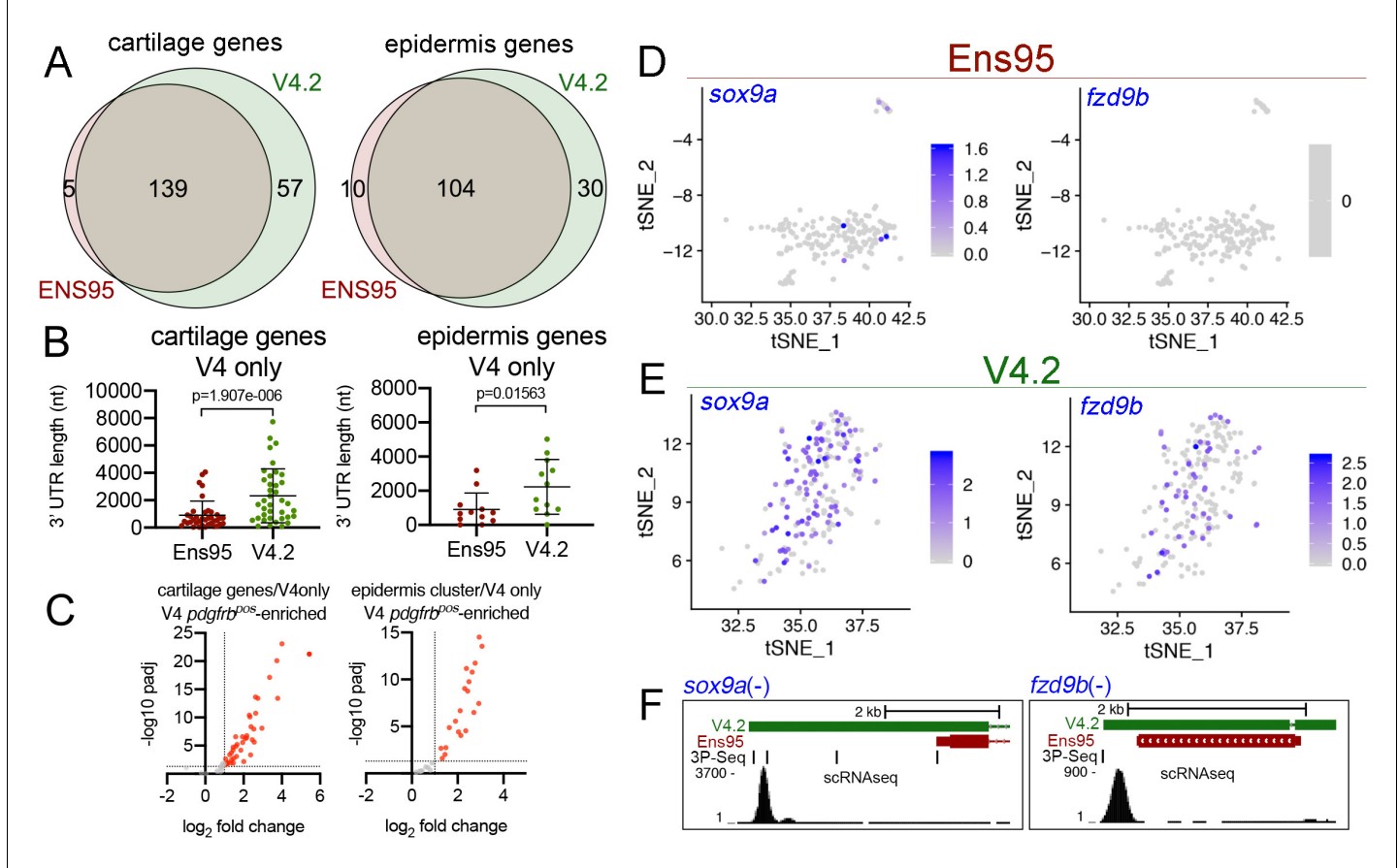

**Figure 5.** The V4.2 annotation increases the quantity of cell type-specific genes identified by scRNA-seq. (**A**) Venn diagrams illustrating intersection by common LL gene ID of genes enriched in *mia*-positive cartilage cells and *and1*-positive epidermis cells by both indicated annotations. (**B**) 3′ UTR lengths from Ens95 and V4.2 for reference genes identified as cartilage or epidermis-specific uniquely by V4.2. Data are not normally distributed (Shapiro-Wilks test), comparison by Wilcoxon matched-pairs signed-rank test, p-values are shown. (**C**) Volcano plots showing cartilage- or epidermis-specific genes identified selectively by V4.2 with corresponding values from bulk RNA-seq comparison of $pdgfrb^{pos}$ and $pdgfrb^{neg}$ cells. Red indicates $\log_2$ fold change >1 ($pdgfrb^{pos}/pdgfrb^{neg}$), padj <0.05. (**D, E**) tSNE plots of the *mia*-positive cartilage cluster showing expression levels of *sox9a* and *fzd9b* using scRNA-seq data quantified using (**D**) Ens95 and (**E**) V4.2. Legends indicate log-transformed and normalized expression levels per cell. (**F**) UCSC browser image of *sox9a* and *fzd9b* loci showing 3′ UTR annotations from V4.2 and Ens95. Both loci are located on the negative strand. Mapped read depth from $kdrl^{pos}$ cells on the genome, or assigned to each annotation are indicated, as are 3P-seq features, all of which reside in the same orientation as the genes themselves.

The online version of this article includes the following source data and figure supplement(s) for figure 5:

**Source data 1.** Cartilage-specific genes identified by scRNA-seq.

**Source data 2.** Epidermis-specific genes identified by scRNA-seq.

**Figure supplement 1.** Improved detection of cartilage and epidermis genes in scRNA-seq data using V4.2.

Accordingly, both *sox9a* and *fzd9b* showed longer 3' UTR annotations in V4.2 compared to Ens95 (*Figure 5F*). Similar results were noted for selected genes in the V4.2 epidermis cluster (*Figure 5—figure supplement 1C*, *Figure 5—source data 2*). In all of these examples, the extended 3' UTR annotations in V4.2 were supported by 3P-seq features and would be expected to capture the majority of reads from scRNA-seq. Notably, the 3' UTR annotated in V4.2 for *fzd9b*, which is also supported by a 3P-Seq feature, is only 350 bp longer than that in Ens95 (*Figure 5F*). Thus, even a modest increase in 3' UTR length can have a significant impact on gene detection, especially in the case for 10x Genomics libraries where short reads are anchored at the 3' end.

We also noted cases where use of the V4.2 annotation could improve clustering resolution over Ens95-quantified scRNA-seq data. For example, we identified several groups of blood cells, which included cells within two clusters expressing leukocyte-specific genes (e.g. *coronin, actin-binding protein, 1A* [*coro1a*] and *Rac family small GTPase 2* [*rac2*]; *Jones et al., 2013*; *Song et al., 2004*; *Tell et al., 2012*; *Figure 6A,B*) and erythroid cells expressing hemoglobin (*hbae1.1*; *Galloway et al., 2005*; *Oates et al., 1999*; *Figure 6A–C*). The two leukocyte clusters could be further distinguished by expression of *src-like adapter 2* (*sla2*, *Figure 6D*), which is expressed in T and B cells in mammals (*Pandey et al., 2002*). Interestingly, the *sla2*-positive (*sla2^pos^*) leukocyte cluster was split on the tSNE plot when using Ens95-quantified reads, such that some cells were more closely associated with the erythroid cluster (denoted by two circles connected by a line in *Figure 6A,B and G*). Despite their assignment to the *sla2^pos^* leukocyte cluster, these separate cells failed to express *sla2* and distinctly expressed *hbae1.1*, similar to cells in the adjacent erythroid cluster (*Figure 6C,D*). This subset of *sla2^pos^* cells also did not express leukocyte-specific genes, but shared expression of *v-myb avian myeloblastosis viral oncogene homolog* (*myb*) and *proliferating cell nuclear antigen* (*pcna*), suggestive of a common proliferative or progenitor-like state (*Figure 6E*; *Liu et al., 2017*; *Soza-Ried et al., 2010*). Using reads quantified with the V4.2 annotation, we identified a group of cells that were similarly located adjacent to, but separate from an erythroid cluster. However, in this case, the erythroid-adjacent cells were also classified as separate from *sla2^pos^*-leukocytes (*Figure 6F*). Mapping of cell barcodes from this V4.2 cluster onto cells clustered using Ens95-quantified reads revealed that these are the same cells clustered with, but split from, *sla2^pos^* leukocytes (*Figure 6G*). This suggested that these cells were mis-classified during clustering using Ens95-quantified data. Based on the expression of *hbae1.1* in these cells (*Figure 6C*) and analysis presented below using V4.2, we refer to this cluster as 'erythroid-2'.

To identify a potential source for distinct cell clustering using V4.2, we intersected cluster-specific genes from the Ens95 and V4.2 *sla2^pos^* leukocytes, along with those from V4.2 erythroid-2 cells. Following intersection based on gene symbol, only a small number of genes were common to all three clusters, including *pcna* and *myb*, which were noted above (*Figure 6E,H*, *Figure 6—source data 1*). Similarly, erythroid markers, such as *hbae1.1*, were shared between the erythroid-2 V4.2 and *sla2^pos^*-leukocyte Ens95 clusters, consistent with the latter being a likely mix of the two cell populations (*Figure 6H*). Importantly, the erythroid-2 cluster displayed many more unique genes than those that were shared with the *sla2^pos^* Ens95 cluster. These included known erythroid-expressed genes such as *solute carrier family 25 member 37* (*slc25a37*), which encodes the Mitoferrin iron transporter, *transferrin receptor 1a* (*tfr1a*), and *integrin alpha 4* (*itga4*), which is expressed in hematopoietic progenitor cells (*Li et al., 2018*; *Liao et al., 2000*; *Shaw et al., 2006*; *Wingert et al., 2004*). Expression of these genes was generally low and detected in very few cells in any of these clusters when using the Ens95 annotation, but all were readily detected with V4.2 (*Figure 7A*). Notably, all of these genes appear preferentially expressed in the erythroid-2 cluster (*Figure 7A*). Furthermore, V4.2 shows annotated 3' UTRs for these genes that are much longer than Ens95 and supported by the concordance of 3P-Seq features (*Figure 7B*). Taken together, these observations suggest that increased 3'UTR coverage and improved cell type-specific gene detection using the V4.2 annotation can contribute to more accurate clustering of cells from scRNA-seq data than existing annotations.

## Discussion

In this study, we describe an improved transcriptome annotation for the zebrafish genome. We document several discrepancies and deficiencies in the existing Ensembl and RefSeq annotations, including incomplete 3' UTR annotations and missing gene models. Our new transcriptome annotation addresses many of these issues and performs better than existing annotations for the analysis of

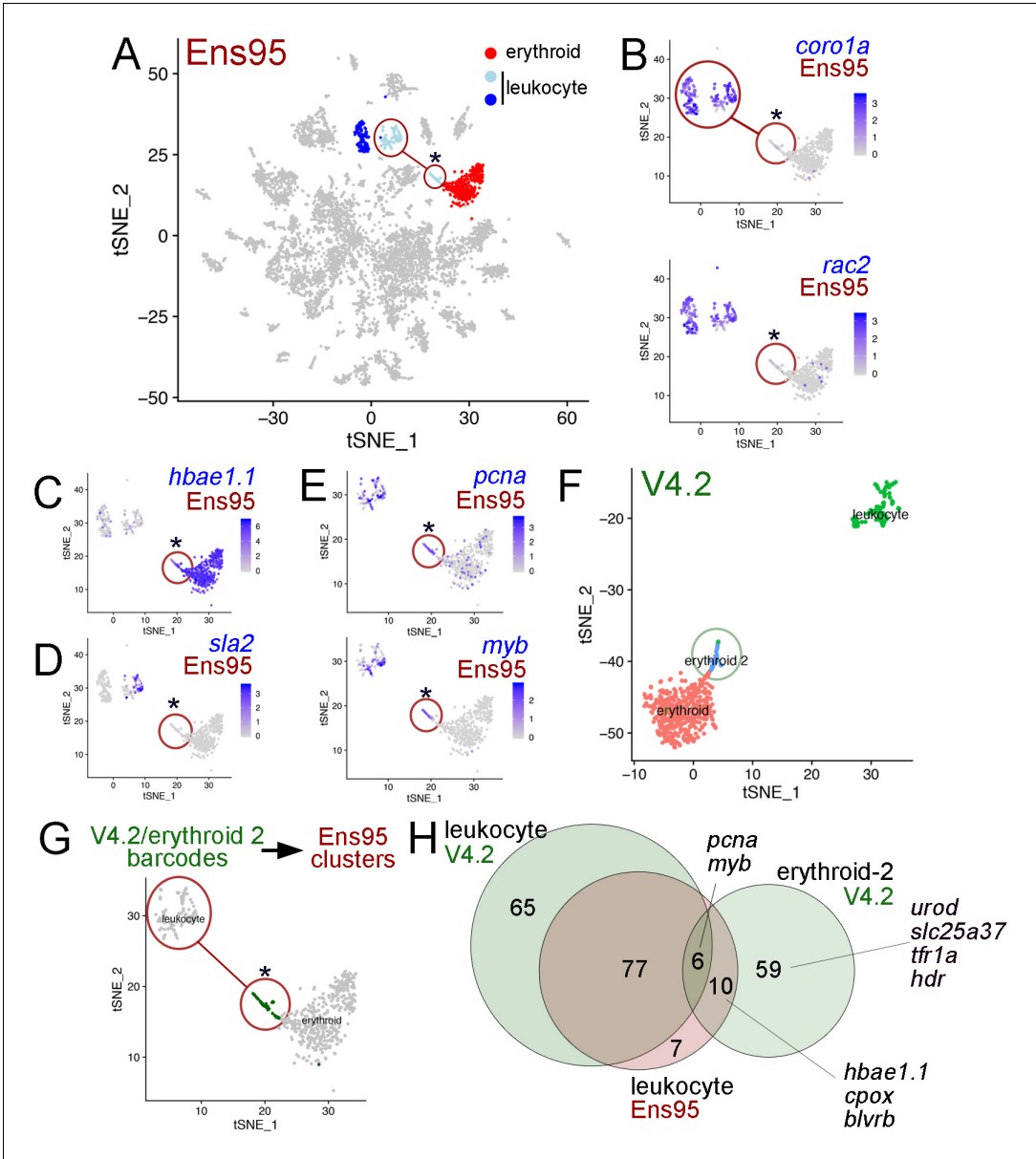

**Figure 6.** The V4.2 annotation improves cluster resolution in scRNA-seq data. (**A**) tSNE plot of all clusters from Ensembl (Ens95)-quantified scRNA-seq of zebrafish embryos at 5 days post fertilization with one erythroid and two leukocyte clusters indicated. (**A–E, G**) Circled clusters denote *sla2$^{pos}$* leukocyte cells. Asterisk marks cells spuriously assigned to *sla2$^{pos}$* leukocytes, but identified as erythroid two when using the V4.2 annotation. (**B–E**) tSNE plots showing only *sla2$^{pos}$* leukocyte (circled) and erythroid clusters with expression of (**B**) known leukocyte markers, *coro1a*, and *rac2*, (**C**) erythroid marker *hbae1.1*, and (**D**) *sla2*. (**E**) tSNE plots showing expression of *pcna* and *myb* in *sla2$^{pos}$* and erythrocyte clusters. (**B–E**) Legends indicate log-transformed and normalized expression values. (**F**) Erythroid and *sla2$^{pos}$* (leukocyte) clusters identified using the same scRNA-seq dataset as in (**A**) but quantified using V4.2. A unique erythroid-like cluster (erythroid 2) is circled. (**G**) tSNE plot of erythroid and *sla2$^{pos}$* leukocyte clusters identified with Ens95 quantified data. Dark green denotes cells with identical barcodes as those in the erythroid two cluster identified using V4.2 shown in (**F**). (**H**) Three-way Venn diagram intersecting cluster-specific genes of indicated clusters by gene symbol. Selected genes are shown.

The online version of this article includes the following source data for figure 6:

**Source data 1.** List of genes from 3-way Venn diagram output shown in **Figure 6H**.

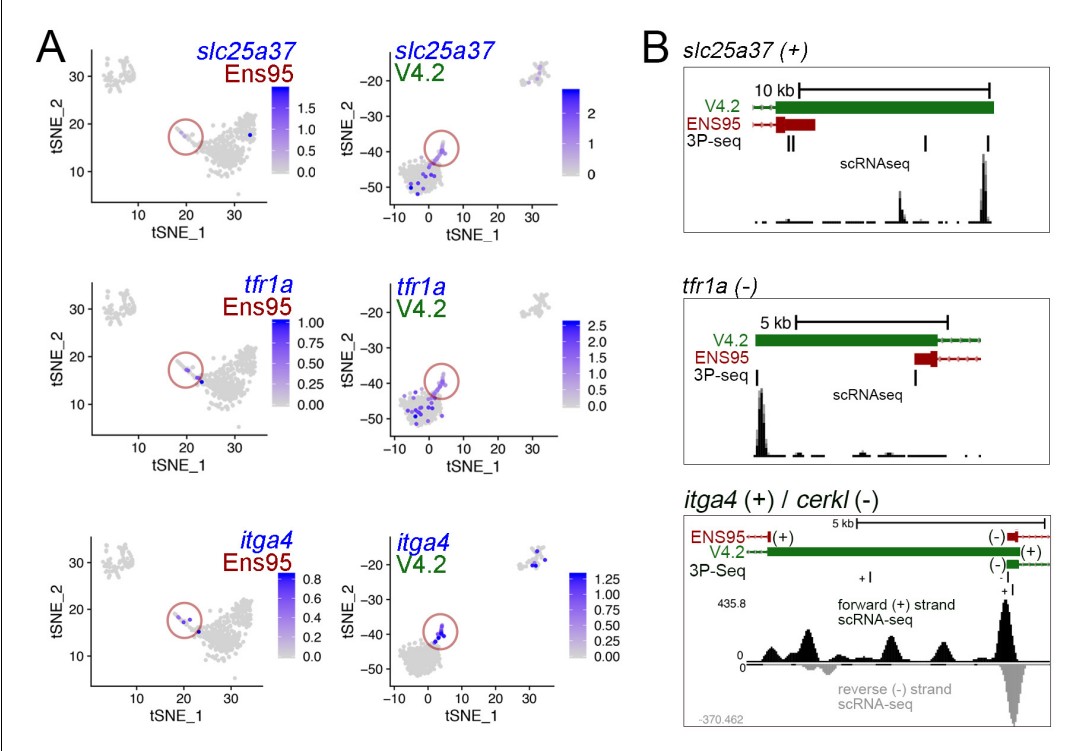

**Figure 7.** Improved detection of erythroid markers in V4.2. (**A**) tSNE plots of blood cell clusters from Ens95- and V4.2-quantified data showing expression of *slc25a37*, *tfr1a*, and *itga4*. Erythroid 2 cells spuriously identified as leukocyte using Ens95 quantification are circled. Legends indicate log-transformed and normalized expression values. (**B**) 3' UTR annotations from Ens95 and V4.2 for *slc25a37* (on the positive strand), *tfra1a* (on the negative strand), and *itga4* (positive strand). The 3' end of the *cerkl* gene on the negative strand overlaps the *itga4* 3' UTR. Therefore, in this case we show read densities split based on strand.

both bulk and single-cell RNA-seq datasets, especially in cases where RNA-seq libraries exhibit reads biased toward the 3' ends of transcripts. Annotation discrepancies proved to be less noticeable when analyzing RNA-seq libraries constructed using a random primed method. However, we noted failure to detect important functional genes and well-known lineage markers even in this case. This issue was especially noticeable in the Ensembl transcriptome annotation, where nearly 20% of all coding genes lack an annotated 3' UTR. Moreover, we also show that integration of bulk RNA-seq and scRNA-seq from common cell types is more consistent when both datasets are mapped to the same annotation. Thus, similar approaches to integrate bulk RNA-Seq from random-primed libraries with scRNA-seq would still benefit from mapping both datasets on an annotation with more accurate 3' UTR models. For these reasons, we believe that our improved annotation will be a valuable tool for zebrafish researchers to apply in the context of their ongoing and planned RNA-seq experiments.

In the course of our previous efforts in applying RNA-seq, we had generated a custom annotation by simply extending the existing Ensembl UTR annotations by 5 kb (previously only described as a 'custom' annotation in the Materials and methods section of *Quillien et al., 2017*). However, we found that this could lead to several secondary issues regarding RNA-seq read and gene assignment, especially in cases where an extension merged with an adjacent gene on the same strand. Indeed, as we find in this study, this is currently an issue with several loci in the recent Ensembl annotations, leading to mis-quantification of RNA-seq data, aberrant assignment of gene information, and apparent loss of genes (for examples see *Figure 3—figure supplement 1A–C*). We have also found that some Ensembl genes are missing terminal exons, such that a simple 3' extension could lead to quantification of otherwise spurious intronic RNA-seq reads from pre-mRNA (data not shown). In addition, several hundred genes from the reference gene set we generated here possess 3' UTRs in the V4 annotation that are more than 5 kb longer than existing Ensembl or RefSeq

annotations. Therefore, we reasoned that the incorporation of existing high-quality RNA-seq datasets would provide an evidence-guided approach to extend 3′ UTR annotations. Indeed, our new annotation exhibits extended 3′ UTR models for a large proportion of existing genes and improves performance for RNA-seq analysis. Moreover, transcript diversity is drastically increased in our annotation, with evidence of many more splice isoforms not evident in Ensembl or RefSeq. While we have not comprehensively addressed the depth of isoform diversity in this study, we believe that our improved annotation will prove useful for better characterizing splice isoform and differential 3′ UTR usage in the zebrafish. Future studies that take advantage of long-read sequencing, along with additional 5′ and 3′ anchored sequencing, will prove helpful at further verifying and improving the transcript models predicted in our annotation.

The increased application of scRNA-seq for molecular analysis in the zebrafish model underscores the need for a transcriptome annotation with accurate 3′ UTR annotations. This is especially important given the reliance on 3′ end sequencing in most scRNA-seq studies. As we show, a 3′ UTR annotation missing just a few hundred nucleotides can lead to drastic differences in gene expression. With the limited sequence space captured in most scRNA-seq libraries, along with the large proportion of incomplete 3′ UTRs in existing zebrafish annotations, many previous studies may be missing a large number of cell type-specific genes. This issue subsequently has a direct impact on cell clustering. In our example, we show how improvements in 3′ UTR annotation lead to a detection of important erythroid-specific genes that guide more accurate assignment of distinct, but related, cell types. Indeed, scRNA-seq analysis with the V4.2 annotation increased the number of clusters called by Ens95 by fifteen percent, suggesting there are numerous examples where improved gene quantification likely led to more refined cell clustering. Whether the V4.2 annotation might also provide similar improvements with scRNA-seq libraries anchored at the 5′ end is not known. Anecdotally, we have noted several genes where V4.2 shows an extended 5′ UTR annotation when compared to RefSeq and Ensembl (e.g. *fzd9b*, see *Figure 5F*). However, we have not quantified whether this observation holds across most of the transcriptome. With the availability of cap analysis of gene expression (CAGE) datasets that define the transcriptional start sites of genes in zebrafish (*Haberle et al., 2014*), it would be possible in future studies to assess how complete the 5′ ends of gene models might be across the different transcriptome annotations.

The use of our annotation for read quantification can improve RNA-seq interpretation through a more accurate estimation of gene expression. However, there remain issues to be improved addressed in future versions. Despite improvements, our annotation is likely limited by the use of short-read sequencing libraries. For example, in cases where genes encode multiple transcript isoforms with both variable 3′ UTR length and exon usage, it is not possible to accurately assign which splice isoforms have particular 3′ UTRs. We would note that this same deficiency applies to Ensembl and RefSeq annotations when quantifying data from RNA-seq libraries generated through short-read deep sequencing. We also noted that some predicted XLOC loci removed from V4.2 were located immediately adjacent and downstream of existing or extended 3′ UTR annotations, suggesting that they were likely longer 3′ UTRs (data not shown). The prevalence of repetitive elements in these 3′ UTR sequences may preclude dense enough mapping of short-read sequences to connect these downstream annotations. As noted above, future integration of our annotation with long-read sequencing will likely further improve both isoform assignment and 3′ UTR models. Likewise, 3′-anchored sequencing efforts, along with more rigorous computational analysis could also better define differential polyadenylation sites and allow improved characterization of differential 3′ UTR usage. We would also point out that the RNA samples we used for reconstructing the V4 transcriptome were limited to embryonic and early larval stages. As such, genes restricted to late larval and adult stages may still have a preponderance of incomplete 3′ UTRs. There are also likely to be some remaining nomenclature issues, particularly in the case of duplicated genes where official name designations have not yet been assigned, or in cases where a gene is split between remaining non-chromosomal contigs. Despite these remaining issues, the current iteration of our transcriptome annotation represents an improvement over existing annotations and will provide a valuable resource for the zebrafish community.

# Materials and methods

**Key resources table**

| Reagent type (species) or resource | Designation | Source or reference | Identifiers | Additional information |
|---|---|---|---|---|
| Genetic reagent (*Danio rerio*) | *Tg(fli1a:egfp)^y1* | *Lawson and Weinstein, 2002* | RRID:ZFIN_ZDB-ALT-011017-8 | |
| Genetic reagent (*Danio rerio*) | *TgBAC(pdgfrb:citrine)^s1010* | *Vanhollebeke et al., 2015* | RRID:ZFIN_ZDB-ALT-150922-2 | |
| Antibody | anti-NR2F2 NR2F2 D16C4, rabbit monoclonal | Cell Signaling Technology | Cat #6434; RRID:AB_11220428 | used at 1:100 |
| Antibody | Goat anti-Rabbit IgG (H+L) Cross-Adsorbed Secondary Antibody, Cyanine3-conjugated | Thermo Fisher Scientific | Cat #A10520; RRID:AB_2534029 | used at 1:1000 |
| Software, algorithm | DolphinNext | *Yukselen et al., 2020*; https://github.com/UMMS-Biocore/dolphinnext | | |
| Software, algorithm | In-house bulk RNA-seq pipeline | https://github.com/UMMS-Biocore/rnaseq | | |
| Software, algorithm | DEBrowser | *Kucukural et al., 2019*; https://github.com/UMMS-Biocore/debrowser | | |
| Software, algorithm | Intervene | *Khan and Mathelier, 2017* https://intervene.readthedocs.io/en/latest/install.html | | |
| Software, algorithm | gffcompare | *Pertea and Pertea, 2020*; https://github.com/gpertea/gffcompare | | |
| Software, algorithm | GenomicFeatures | *Lawrence et al., 2013*; https://bioconductor.org/packages/release/bioc/html/GenomicFeatures.html | RRID:SCR_016960 | |
| Software, algorithm | bedtools | *Quinlan and Hall, 2010*; https://bedtools.readthedocs.io/en/latest/content/installation.html | RRID:SCR_006646 | |
| Software, algorithm | GSNAP | *Wu and Nacu, 2010*; http://research-pub.gene.com/gmap/src/gmap-gsnap-2019-09-12.tar.gz | RRID:SCR_005483 | |
| Software, algorithm | Cufflinks/Cuffmerge | *Trapnell et al., 2010*; http://cole-trapnell-lab.github.io/cufflinks/cuffmerge/ | RRID:SCR_014597 | |
| Software, algorithm | Cell Ranger | 10x Genomics; https://support.10xgenomics.com/single-cell-gene-expression/software/downloads/latest | RRID:SCR_017344 | |
| Software, algorithm | Seurat | *Stuart et al., 2019*; https://satijalab.org/seurat/install.html | RRID:SCR_007322 | |

## Zebrafish lines and maintenance

All studies were performed under the auspices of animal protocols approved by the University of Massachusetts Medical School Institutional Animal Care and Use Committee. The *Tg(fli1a:egfp)^y1* and *TgBAC(pdgfrb:citrine)^s1010* trangenic lines have been described elsewhere (*Lawson and Weinstein, 2002*; *Vanhollebeke et al., 2015*).

## RNA-seq sample preparation

For *pdgfrb:citrine* analysis, citrine-positive and -negative cells were obtained by fluorescent activated cell sorting (FACS) from dissociated *TgBAC(pdgfrb:citrine)$^{s1010}$* embryos at five dpf and used to generate RNA-seq libraries as previously described (*Quillien et al., 2017*; *Whitesell et al., 2019*). To isolate venous endothelial cells, we applied a modification of the MARIS protocol that we previously adapted to zebrafish (*Hrvatin et al., 2014*; *Quillien et al., 2017*). Briefly, dechorionated and deyolked *Tg(fli1a:egfp)$^{y1}$* embryos at 38 hr post-fertilization were dissociated by immersing in 1 x TrypLE Express for 15 min at 28.5˚C. All subsequent steps took place at 4˚C. Cells were filtered through a 70 μm cell strainer, centrifuged at 3,000 rpm for 3 min, and resuspended in 4% Paraformaldehyde/0.1% Saponin in phosphate buffer saline (PBS) for 30 min with rotation. Cells were washed twice (0.25% BSA/0.1% Saponin in PBS with RNAsinPlus (280 U/mL)) to remove fixative and then incubated with anti-NR2F2 (D16C4) Rabbit mAB (#6434, Cell Signaling) at 1:100 in blocking buffer (1% BSA/1% Saponin in PBS with RNAsinPlus (400 U/mL)) with rotation for 15 min. Cells were washed for 5 min twice and incubated with goat anti-rabbit IgG(H+L)-Cy3 (A10520, Invitrogen) at 1:1000 in blocking buffer for 1 hr. Cells were then washed twice and resuspended in wash buffer. Fluorescence-activated cell sorting (FACS) was performed by the UMassMed Flow Cytometry Core to separately isolate Nr2f2/EGFP double-positive (red/green) and Nr2f2-negative/EGFP-positive (green) cells. Total RNA was purified from double- and single-positive cells using RecoverAll Total Nucleic Acid Isolation Kit (AM1975, Ambion), then concentrated using RNA Clean and Concentrator-5 Kit (11–325, Zymo Research). After the elimination of ribosomal RNAs using mammalian Ribo-Gone Kit (634846, Clontech), RNAseq libraries were prepared using SMARTer Stranded RNA-Seq Kit (634836, Clontech) and deep sequencing performed on a HiSeq2000 (UMass Med Deep Sequencing Core Facility). Libraries were constructed from Nr2f2/EGFP-positive and EGFP-positive cells isolated from clutches of embryos generated on three different days.

## Bulk RNA-seq analysis

We ran mapping and quantification pipelines described below in the DolphinNext environment (https://github.com/UMMS-Biocore/dolphinnext; *Yukselen et al., 2020*) using the Massachusetts Green High Performance Computing Cluster (GHPCC). Triplicate samples of paired-end reads from *kdrl*-positive and *kdrl*-negative cells were downloaded from GEO (GSE119718). For bulk RNA-seq we utilized a standardized pipeline in the DolphinNext environment (https://github.com/UMMS-Biocore/rnaseq, *Yukselen et al., 2020*). Briefly, reads were mapped to GRCz11 using STAR (*Dobin et al., 2013*) with default parameters to generate BAM files, which were subsequently used for transcript quantification with RSEM (*Li and Dewey, 2011*) using Ensembl, version 95, RefSeq (GCF_000002035.6_GRCz11), or V4.3 transcript annotations. We identified differentially expressed genes using DESeq2 with gene-mapped expected count files from RSEM. DESeq2 was run using the DEBrowser interface (*Kucukural et al., 2019*). We applied the Low count filtering method to remove features with counts per million of less than 10 in at least three samples and median ratio normalization was applied in all cases. The dispersion was estimated using a parametric fit and hypothesis testing performed using the likelihood ratio test. p-values obtained by the Wald test were adjusted for multiple testing using the Benjamini and Hochberg method (*Benjamini and Hochberg, 1995*). Differentially expressed genes were defined as those with log$_2$ fold change less than $-1$ or greater than one and adjusted p-value less than 0.05. We ran the same pipeline with identical parameters for each annotation (RefSeq, Ens95, or V4.3). Volcano plots were generated using DESeq2 in the DEBrowser interface.

## Annotation comparisons

To compare RNA-seq analyses from Ens95 and RefSeq, we downloaded matching NCBI IDs for Ensembl-annotated genes using Biomart from the Ens95 archive (http://jan2019.archive.ensembl.org/Danio_rerio/Info/Index). Our gene-level DEbrowser output listed gene symbol with all associated transcripts. Since a number of Ens95 and RefSeq genes have identical gene symbol annotations, we incorporated matching Ens95 gene and NCBI IDs using a representative transcript ID for each gene to ensure correct matches. We then applied Intervene (*Khan and Mathelier, 2017*) for intersections based on common NCBI ID and generation of Venn diagrams. Proportional Venn diagrams were generated using eulerr (*Larsson, 2020*, https://github.com/jolars/eulerr.co). A similar

approach was applied for comparison of RNA-seq outputs from V4.2 and V.3, except that LL gene ID was used. All corresponding NCBI, Ensembl, and LL gene IDs for RNA-seq output are included in the corresponding source data tables. To identify matching and missing genes between different annotations, we applied gffcompare (*Pertea and Pertea, 2020*), which provides a coordinate-based comparison of transcript intron/exon structure using GTF files as input. A GTF for RefSeq (GCF_000002035.6_GRCz11) was downloaded from the UCSC Table browser, while Ens95 was downloaded from the Ensembl FTP site (ftp.ensembl.org:21/pub/release-95/gtf/danio_rerio/Danio_rerio.GRCz11.95.gtf.gz). Before running gffcompare, Ensembl-annotated GTF files were modified to match UCSC-based chromosome annotation, while 'alt' chromosome variants in RefSeq were removed. Each pairwise comparison was run with default settings using each annotation as a reference to identify relative missing and matching transcript models. To confirm that missing transcripts represented actual missing genes, reference genomes in each case were subsequently re-searched for matching gene IDs corresponding to each missing transcript. We subsequently surveyed between 10 to 20 missing genes manually through individual visualization of their loci on the UCSC genome browser to confirm the results. Matched transcript outputs from gffcompare, along with missing gene lists can be found in the associated source data files.

## Comparison of 3' UTR coverage between annotations

To assess the presence or absence of 3' UTR models in coding genes in Ens95 and RefSeq, we relied on the annotation of these features in column 3 of each respective gene transfer format file (GTF). Each feature was extracted from the respective GTF, collapsed to novel gene ID to avoid spuriously counting alternatively spliced multi-exon 3' UTRs, and counted. To compare 3' UTR length, we chose to generate a reference gene set with a single representative transcript per gene. This approach reduces the potential for spuriously matching or assigning differentially spliced or alternative 3' UTR isoforms across annotations. All 3' UTR lengths were obtained from each annotation by using GenomicFeatures in R (*Lawrence et al., 2013*) with GTFs for each annotation containing only the 3' UTR annotated entries. We next extracted all terminal exons from the Ens95 GTF annotated as a 3' UTR. For each Ens95 gene, we then identified and chose a single transcript with the longest 3' UTR as representative. To accurately match a RefSeq transcript to each representative Ens95 transcript, we first identified and extracted the longest terminal 3' UTR exon for each RefSeq transcript. In cases where there were identical overlapping transcript coordinates, these were collapsed into a single entry and arbitrarily assigned one of the duplicate transcript IDs. Matching transcripts from RefSeq were integrated using bedtools by intersecting on the last exon start coordinates on the same strand (i.e. start coordinate on the positive stand, end coordinate on the minus strand). To incorporate V4.3 genes, we identified the longest terminal exon for each gene, collapsed duplicates, and assigned an arbitrary transcript ID as above for RefSeq entries. V4.3 transcripts matching the reference gene were then incorporated using bedtools to intersect on exon start coordinates on the same strand as above. V4.3 3' UTR lengths were then calculated based on the differences between RefSeq or Ens95 (see *Figure 2—source data 2*). To visualize 3P-Seq location, we downloaded BED files from GEO (GSE37453, *Ulitsky et al., 2012*) and merged these into a single file after filtering for expression level. We then sequentially lifted these coordinates over from Zv9 to GRCz10 and then to GRCz11 using the UCSC browser interface (http://genome.ucsc.edu/cgi-bin/hgLiftOver). For comparison of gene-level expression between annotations, we imported values from DESeq2 output for indicated dataset mapped to each respective annotation using the appropriate matched ID. Further details are available upon request.

## Assembly of an improved zebrafish transcriptome annotation

We downloaded 75 bp Paired-end stranded RNA-seq reads from embryos staged at 28hpf, 2dpf, 5dpf, bud, dome, and shield from GEO (Accession no.: GSE32898, *Pauli et al., 2012*). Reads were trimmed with Trimmomatic (adaptor removal, headcrop, and crop to retain position 10–50 bp only along with default quality trimming) and reads shorter than 32 bp after trimming were discarded. Trimmed reads were aligned to *Danio rerio* genome GRCz11 with GSNAP (*Wu and Nacu, 2010*) allowing 5% mismatches. Our anecdotal observations suggested a prevalence of incomplete and spurious partial transcripts in Ensembl. Therefore, the alignment was guided by splicing sites extracted from refseq_mRNA (GCF_000002035.6_GRCz11). We filtered alignment results so that

only properly paired reads with MQ $\geq$30 and more than 35 bp of mapped regions were retained. An in-house perl script was used to add strand information and format the filtered alignment results (*Source code 2*). We used mapped reads to assemble transcript and gene models using Cufflinks and Cuffmerge (*Trapnell et al., 2010*). Preliminary versions of our transcriptome annotation (v1, v2) were used to identify parameters that optimally captured isoform diversity and 3' UTR lengths. Version 3.1.2 (v3.1.2) used Cufflinks with the following parameters: -F 0.2 j 0.3 -a 0.0001 -A 0.4 `–min-frags-per-transfrag 20 –overhang-tolerance 20 –overlap-radius` 50 -`–3-overhang-tolerance` 50 to assemble transcripts for samples from each embryonic stage. Cufflinks assembly for v3.1.2 was guided (using the -g argument in Cufflinks) by the most recently available zebrafish RefSeq annotation (GCF_000002035.6_GRCz11). We next integrated trans-fragments assembled by Cufflinks for each stage using Cuffmerge. We chose to use Ensembl annotations (version 95) to guide the integration of the transcript models generated above using Cuffmerge. At this step, we allowed low abundance transcripts to be retained in the final annotation ('`–min-isoform-fraction` 0.00'). Initial gene nomenclature assignments were made using v3.1.2. For this purpose, all relevant matching gene information was extracted from the GTF output from Cuffmerge for v3.1.2 and used to assign gene information to transcript ID (TCONS) generated by Cufflinks. At this point, over 1000 gene symbol conflicts were identified by comparing assigned gene names for comparing RefSeq and Ens95 transcript IDs to those assigned by Cufflinks. In many cases, these were due to spurious fusion transcripts linking adjacent genes. Therefore, we sought to remove these using a customized Python script (*Source code 3*). We also manually removed non-miRNA small RNAs. Given the nomenclature conflicts noted above, we utilized gffcompare to accurately import matched gene identifiers from RefSeq and a more recent Ensembl annotation (version 99, released January 2020). In cases where a perfect transcript match was not available, we subsequently compared gene coordinates to further confirm overlap; we include a qualitative description for each entry in this case for RefSeq and Ens99 matches in the V4.3 gene information file. In the course of investigating the nature of the nomenclature conflicts noted above, we also manually corrected several incorrect gene names assigned by Ensembl that resulted from persistent overlapping transcripts between two separate genes that evaded filtering noted above, causing a single gene ID to be assigned to two genes (see main text and *Figure 3—source data 2*). In these cases, spurious overlapping transcripts were removed to give V4.2. To identify any remaining Ensembl or RefSeq genes missing from V4.2, we performed pairwise comparisons with each annotation using gffcompare. The resulting missing genes were then manually added to give V4.3. Since our annotation comprises genes missing from both Ensembl and RefSeq annotations, as well as novel genes (see below), we assigned a unique identifier (prefix 'LL' followed by an arbitrary number) to each entry, which can be used as a common link for the relation between the gene information file and the resulting output from the GTF following its application in RNA-seq analysis. The GTF and gene information files for V4.2 and V4.3 are available at http://zf-transcriptome.umassmed.edu.

## Annotation of novel loci

Our annotation yielded just over 6000 novel loci with no initial match to Ensembl or RefSeq genes. These are annotated using gene abbreviations with an 'XLOC' prefix assigned by cufflinks. To investigate whether these represent new genes, we took several steps. First, we used transcript sequences from each locus in blastx searches against zebrafish and human proteins. Second, we determined whether any overlapped previously long non-coding RNAs identified in previous studies. For this purpose, we used an integrated annotation published elsewhere that includes lincRNAs not found in Ensembl or RefSeq (*Hu et al., 2018*). This annotation was converted to a standard BED format, lifted from Zv9 to GRCz10, then to GRCz11 and used to generate a GTF file. We used gffcompare with the lincRNA GTF file and a parallel file containing only XLOC V4.2 annotations to identify matching or overlapping transcripts. Finally, we determined exon numbers for each XLOC entry. In V4.2, all XLOC entries were maintained in the associated GTF and gene information files. For V4.3, we retained XLOCs that matched one of the following criteria: 1. exhibited exact, moderate, or high-confidence blast hits with a zebrafish or human protein, 2. exhibited matched or overlapping intron-exon structure with the lincRNAs noted above, or 3. comprised more than one exon. We also identified approximately 200 loci that exhibited complete overlap with existing RefSeq-annotated transcripts already present in V4.2. These redundant XLOC loci were also removed in the update to V4.3.

## Single-cell RNA-seq analysis

We downloaded scRNA-seq reads from whole zebrafish embryos at five dpf from the Sequence Read Archive (SRR10095965, *Farnsworth et al., 2020*). Paired-end reads were split from this file using the SRA toolkit (https://github.com/ncbi/sra-tools; *SRA Toolkit Development Team, 2020*). We used Cell Ranger (10x Genomics, v3.1.0) in the DolphinNext environment (see above) for read mapping, filtering, and counting using default parameters. We ran Cell Ranger separately with the same set of reads and parameters for Ens95 and V4.2. Cell Ranger output files (barcodes, genes, and matrix files) were loaded into Seurat (v3.1, *Stuart et al., 2019*) for clustering, identification of differential genes, and visualization. All parameters and variables can be found in the accompanying R file (*Source code 1*). Seurat was run in RStudio through a web-based portal on the GHPCC using Open OnDemand (*Hudak et al., 2018*).

## Data access

New raw datasets (*pdgfrb:citrine* and Nr2f2 RNA-seq), and associated processed data files, generated for this manuscript, including the V4.3 GTF, are available at the Gene Expression Omnibus (GSE152759). All other raw datasets were otherwise obtained from GEO. Additional support files are available at http://zf-transcriptome.umassmed.edu.

## Acknowledgements

We thank members of the UMass Med GHPCC group, including Lewis Robbins and Chris Hull for their support. We are also grateful to Yu-Huan Shih and William Frantz for their helpful comments on the manuscript. This work was supported by NIH R35HL140017 and R21NS105654 to N D L and U01HG007910 from NHGRI and National Center for Advancing Translational Sciences grant UL1 TR001453, which supported A K and O Y.

## Additional information

### Funding

| Funder | Grant reference number | Author |
| --- | --- | --- |
| National Heart, Lung, and Blood Institute | R35HL140017 | Nathan D Lawson |
| National Human Genome Research Institute | U01HG007910 | Onur Yukselen Alper Kucukural |
| National Center for Advancing Translational Sciences | UL1TR001453 | Onur Yukselen Alper Kucukural |
| National Institute of Neurological Disorders and Stroke | R21NS105654 | Nathan D Lawson |

The funders had no role in study design, data collection and interpretation, or the decision to submit the work for publication.

### Author contributions

Nathan D Lawson, Conceptualization, Data curation, Formal analysis, Supervision, Funding acquisition, Validation, Investigation, Visualization, Methodology, Writing - original draft, Project administration, Writing - review and editing; Rui Li, Data curation, Software, Formal analysis, Methodology, Writing - review and editing; Masahiro Shin, Ann Grosse, Investigation, Methodology; Onur Yukselen, Lihua Zhu, Resources, Software, Methodology, Writing - review and editing; Oliver A Stone, Resources; Alper Kucukural, Resources, Software, Visualization, Methodology, Writing - review and editing

### Author ORCIDs

Nathan D Lawson (iD) https://orcid.org/0000-0001-7788-9619
Alper Kucukural (iD) http://orcid.org/0000-0001-9983-394X

## Ethics

Animal experimentation: Zebrafish studies were performed in accordance with protocols #A2613 and #A2632 approved by the University of Massachusetts institutional animal care and use committee (IACUC).

## Decision letter and Author response

Decision letter https://doi.org/10.7554/eLife.55792.sa1
Author response https://doi.org/10.7554/eLife.55792.sa2

# Additional files

## Supplementary files

• Source code 1. List of R commands used to run Seurat for clustering of data quantified using Ens95 and V4.2.

• Source code 2. Perl script used to add strand information and filter reads in BAM file output from GSNAP.

• Source code 3. Python script to identify and remove spurious fusion transcripts.

• Source data 1. Zebrafish transcriptome annotation V4.2. Contains genomic annotation file (153.8 MB, md5sum:44c87a2bdd19ccfd9f7cd526f9e21498) and gene information (as tab-delimited file and .xlsx file).

• Source data 2. Zebrafish transcriptome annotation V4.3.1. Contains genomic annotation file (152 MB, md5sum: 19759898187c47edfd9c216162851e31) and gene information (as tab-delimited file and .xlsx file).

• Transparent reporting form

## Data availability

All data generated in this study are available in accompanying source data files. Transcriptome annotation files described in this study are available for download at http://zf-transcriptome.umassmed.edu. Raw and processed RNA-seq data generated in this study are available at GEO (GSE152759).

The following dataset was generated:

| Author(s) | Year | Dataset title | Dataset URL | Database and Identifier |
|---|---|---|---|---|
| Lawson ND | 2020 | Bulk RNA-seq data to assess an improved zebrafish transcriptome annotation | https://www.ncbi.nlm.nih.gov/geo/query/acc.cgi?acc=GSE152759 | NCBI Gene Expression Omnibus, GSE152759 |

The following previously published datasets were used:

| Author(s) | Year | Dataset title | Dataset URL | Database and Identifier |
|---|---|---|---|---|
| Whitesell TR, Chrystal PW, Ryu JR, Munsie N, Grosse A, French CR, Workentine ML, Li R, Zhu LJ, Waskiewicz A, Lehmann OJ, Lawson ND, Childs SJ | 2019 | Morphogenesis and differentiation of embryonic vascular smooth muscle cells in zebrafish | https://www.ncbi.nlm.nih.gov/geo/query/acc.cgi?acc=GSE119718 | NCBI Gene Expression Omnibus, GSE119718 |
| Pauli A, Valen E, Lin MF, Garber M, Vastenhouw NL, Levin JZ, Sandelin A, Rinn JL, Regev A, Schier AF | 2011 | Comprehensive identification of long non-coding RNAs expressed during zebrafish embryogenesis | https://www.ncbi.nlm.nih.gov/geo/query/acc.cgi?acc=GSE32900 | NCBI Gene Expression Omnibus, GSE32900 |

| Ulitsky I, Shkumatava A, Jan CH, Subtelny AO, Koppstein D, Bell G, Sive H, Bartel DP | 2012 | Extensive alternative polyadenylation during zebrafish development | https://www.ncbi.nlm.nih.gov/geo/query/acc.cgi?acc=GSE37453 | NCBI Gene Expression Omnibus, GSE37453 |

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
