## [Decision Letter]

**Acceptance summary:**

This paper demonstrates that current RefSeq and Ensembl gene annotations of the zebrafish genome have discrepancies such as mutually exclusive gene models that can affect the interpretation of transcriptional profiling datasets. Moreover, incomplete annotation of 3' untranslated regions impairs gene expression analysis of datasets from increasingly popular single cell RNA-seq techniques due to their inherent 3' bias. To address these problems the authors have produced a new zebrafish gene annotation that improves detection of cell-type specific transcripts.

**Decision letter after peer review:**

Thank you for sending your article entitled "An improved zebrafish transcriptome annotation for sensitive and comprehensive detection of cell type-specific genes" for peer review at *eLife*. Your article is being evaluated by three peer reviewers, and the evaluation is being overseen by a Reviewing Editor and Didier Stainier as the Senior Editor.

In this manuscript, Lawson et al. observe that some 3' UTRs in the Ensembl and RefSeq zebrafish gene annotations seem incomplete and gene models are inconsistent between the two annotations. The incomplete 3' UTRs affect their ability to identify differentially expressed genes in an RNA-seq dataset from *kdrl^pos^* and *kdrl*^neg^ cell populations. The authors speculate that this is due to the strong 3' bias in read distribution caused by oligo-dT priming. To remedy this, the authors use Cufflinks and Cuffmerge to produce transcript and gene models from a published RNA-seq dataset covering six developmental stages, guided by RefSeq splice sites, and Ensembl annotation. The resulting annotation (called V4.2) has longer 3' UTRs for a subset of genes. The authors show that this increases the number of genes detected as differentially expressed in their RNA-seq dataset. Similarly, mapping a published single cell 10x Genomics RNA-seq dataset against V4.2 moderately increases the number of detected genes and cell clusters in a tSNE plot. The authors conclude that these results demonstrate that their V4.2 annotation is superior to the Ensembl and RefSeq annotations and recommend using V4.2 for bulk and single cell RNA-seq analysis.

The reviewers agree that the current 3' UTR annotation of the zebrafish genome is incomplete and might affect the interpretation of 3' biased transcriptional profiling data. Therefore, an annotation with more accurate 3' UTRs would be a useful resource. However, as it stands, the annotation presented here would create different problems for anyone using it because of the additional (and in this manuscript undiscussed) gene and transcript models that are based solely on one RNA-seq dataset. A Cufflinks transcript model, even if it produces a BLAST hit, is insufficient evidence that it is a bona fide expressed transcript (see below for details). We would therefore ask that the manuscript focus on the improved 3' UTR annotation of Ensembl and/or RefSeq gene models and show that it provides a considerable advancement for RNA-seq analysis. This focus should also be reflected in the title.

Essential revisions:

1) The case for an improvement rests mostly on the fact that a few genes that the authors had expected to change expression in one of their datasets were only detected as differentially expressed (DE) when using their V4.2 annotation instead of Ensembl or RefSeq annotation. The other test case, a single cell RNA-seq re-analysis using V4.2, shows only minor changes. Crucially, there is no evaluation of the gene overlap with the Ensembl or RefSeq based analyses, so some genes might actually be lost. The bulk RNA-seq dataset in this manuscript is unusual as it uses oligo-dT priming instead of the more common random priming used in, for example, the TruSeq library preparation kits. The resulting 3' bias is extreme. For example, in Figure 1C the read depth drops by 99% from 2750 at the 3' UTR end of the gene to 28 in the coding part of that exon. To demonstrate that V4.2 represents a significant improvement over current annotation, an objective analysis of several datasets is required, including random-primed RNA-seq data. If the advantage is limited to 3' end biased data, this should be shown and discussed.

2) The authors state that new transcripts and genes are present in their V4.2 annotation that are missing in both Ensembl and RefSeq. This is based on transcript and gene model output from cufflinks. Some of these models have BLAST hits, which the authors interpret as evidence that these are bona fide expressed transcripts. This is incorrect. Reads can mis-map to processed pseudogenes and other gene remnants and thus look like separate expressed genes. It is therefore not surprising that some of those have BLAST hits. Of the 6562 genes with an XLOC name in v4.2.1.gtf, 4714 are single exon genes. This indicates that the vast majority are indeed likely to be processed pseudogenes. If these gene models were to be used to map against, they would mop up reads from coding genes and distort results. Taken together these models have not passed the many filtering and validation steps using additional evidence (cDNA, cross species comparisons etc) that Ensembl and RefSeq employ. To avoid over-interpretation of these unverified gene models they should be provided separately from those that have Ensembl and/or RefSeq identifiers. The text needs to acknowledge the limitation of these gene models. Claims of missing genes should be made much more cautiously unless independent evidence beyond BLAST hits is provided.

3) The annotation needs to include existing Ensembl and RefSeq identifiers. Gene symbols are notoriously unreliable and not suitable for comparative analyses.

4) The annotation uses one RNA-seq dataset derived from whole embryos and a limited set of developmental stages. It is possible that the modest improvement is partially due to missing tissue- or stage-specific transcripts and their 3' UTRs. The authors should compare the genes detected in a few tissue-specific datasets with V4.2 to demonstrate sufficient overlap or acknowledge the limitation if a substantial number of genes are absent from the used dataset. Alternatively, the authors could run Cufflinks without Ensembl and RefSeq annotation and then compare how many transcripts overlap Ensembl transcripts. Likewise, alternative polyadenylation and differential 3' end use will affect interpretation of gene expression. This should be discussed.

5) The authors need to provide statistics and numbers in the text instead of qualifiers such as "much lower". For example, the authors state "…the overall number of differentially-expressed transcripts identified when using Ensembl was much lower than that found with RefSeq." The difference is 7%. Another example is "increase in the number of median genes". How many? (It's 842 vs. 761.) Likewise, the authors need to provide statistics of their annotation. How many genes are detected, what is the length/ exon number distribution of known transcripts vs. XLOC transcripts etc.

6) A lot of the analysis methodology is missing and no code is provided. "In-house Perl scripts" need to be made available. What were the criteria for "hand-curation"? The same is true for other methods. For example, the intersection analysis of RefSeq and Ens95 is not clear. What does intersection by gene coordinate mean? Is it exactly matching start/end positions or by interval overlap? The discrepancy of 3000 genes is improbably large.

Other required changes and clarifications:

1) In Figure 5A, there are 7 cartilage and 9 epidermis genes unique to Ens95 vs. V4.2. Could the authors comment on this? Are these genes bona fide markers of these cell types or false positives? Will the missing genes in V4.2 be incorporated?

2) Figure 1A, B should be -log.

3) In Figure 2—figure supplement 1, the 6743 genes from RefSeq only in panel A does not correspond to the 6119 genes (2514+3605) analyzed in panel B. Likewise, the 2514 genes from panel B do not correspond to the 2549 genes (2035+514) in panel C. Please correct or explain these discrepancies in gene number.

4) "Comparison of 3' UTRs for matching genes from the two annotations in those latter cases revealed that the overall differences in length were significant (Figure 2C)." Why has a statistical test been performed here? The authors are not randomly sampling from an underlying population of gene models and then measuring the UTR length. They have selected for the ones that are either longer or shorter in Ensembl 95.

5) Figure 3B: If the Ensembl annotation was merged into V4.2, how can V4.2 have shorter UTRs than Ensembl?

6) The counting against the two annotations seems oddly inconsistent. For *slc2a1a* (Figure 1E and Figure 5D) the same exon portion gets 16 reads in Ens95, but only 4 reads in RefSeq.

7) It is not clear what is meant by this sentence in the legend for Figure 2—figure supplement 1: "Total numbers of unique genes in each case are not equivalent due to cases there ("where" I assume?) multiple genes from either or both annotations may intersect within a single interval." How many genes does the overlap method fail to distinguish because of this?

8) "…all of these genes appear preferentially expressed in the erythroid-2 cluster (Figure 7A)". *slc25a37* and *tfr1a* are expressed in more cells at higher levels in the erythroid cluster, not erythroid-2.

9) "…mapped onto the RefSeq annotation". Mapping is to the assembly not the annotation, so it should either say "mapped to GRCz11 and counted against the RefSeq annotation" or just "counted against the RefSeq annotation". Same at : "to genome-mapped or ENSEMBL-mapped reads". The reads aren't mapped to the Ensembl annotation, they are counted against it. This needs changing throughout.

10) Figure 3 legend. What do the bars represent in Figure 3C, E and F, median or mean?

11) Legends for Figures 4-7. What do the colour bars in the t-SNE expression plots represent? It will be some kind of count, but what exactly?

12) It would be interesting to see what the RefSeq and Ensembl comparison looks like for other model organisms (not humans, which has MANE https://www.ncbi.nlm.nih.gov/refseq/MANE/).

13) Figure 5C: There are a lot fewer dots than the Venn diagram would suggest. Are they plotted on top of each other? Jitter might help. Also, what do the genes from the Ens95 counting that don't overlap with the V4.2 ones look like?

14) https://www.umassmed.edu/lawson-lab/transcriptome gives a 404 error.

15) Ensembl not ENSEMBL.

16) Median ration => Median ratio.

17) "In this process, we also manually corrected several incorrect gene names assigned by ENSEMBL that resulted from transcripts overlapping two separate genes, causing a single gene ID to be assigned to two genes." Have these been reported to Ensembl?

18) tRNAscan is mentioned in the Materials and methods, but nowhere else.

19) It would be good if the GTF file contained CDS entries.

20) Figure 2: Venn diagram is incomplete, lacks the non-concordant population

21) Figure 2—figure supplement 1: the Venn diagrams should be size proportionate.

22) Figure 3 and text: reorder bud, shield and dome to dome, shield and bud so as to fully maintain sequence of stages

23) Figure 2B: caption should be V4.2

24) Figure 2C: scales of charts should be fixed to 15k for direct comparison and to demonstrate length difference of 3'UTR distribution between annotations

25) "…only genes with matched gene symbols between ENSEMBL (v95) and RefSeq were maintained" and "only V4.2 genes that matched by gene symbol were incorporated into the reference set". Why do genes which have gene symbol discrepancies between Ensembl and RefSeq were excluded from the reference set? If the genome location and sequence are the same, this may be a common naming issue, rather than a reason to suggest difference in gene/transcript.

26) The new annotation is not currently accessible at the website cited. Will the authors make an effort to add an improved annotation to Ensembl?

27) Figure 6: add consistent labelling of cell types and gene symbols throughout.

---

## [Author Response]

Essential revisions:1) The case for an improvement rests mostly on the fact that a few genes that the authors had expected to change expression in one of their datasets were only detected as differentially expressed (DE) when using their V4.2 annotation instead of Ensembl or RefSeq annotation. The other test case, a single cell RNA-seq re-analysis using V4.2, shows only minor changes. Crucially, there is no evaluation of the gene overlap with the Ensembl or RefSeq based analyses, so some genes might actually be lost. The bulk RNA-seq dataset in this manuscript is unusual as it uses oligo-dT priming instead of the more common random priming used in, for example, the TruSeq library preparation kits. The resulting 3' bias is extreme. For example, in Figure 1C the read depth drops by 99% from 2750 at the 3' UTR end of the gene to 28 in the coding part of that exon. To demonstrate that V4.2 represents a significant improvement over current annotation, an objective analysis of several datasets is required, including random-primed RNA-seq data. If the advantage is limited to 3' end biased data, this should be shown and discussed.

In the revised manuscript, we now provide a more comprehensive analysis of differences between the RefSeq and Ensembl transcriptome annotations and the impact on RNA-seq interpretation. The new analyses include: (1) more detailed comparison of differences and overlap in gene calls across two additional bulk RNA-seq datasets (including one from a library generated through random-priming); (2) use of an improved reference geneset of over 16,000 genes to analyze 3' UTR annotation differences between the two annotations; and (3) a more direct and definitive transcript level comparison of RefSeq and Ensembl annotations, as well as comparison to the new annotations to identify genes from each annotation.

The results support our initial contention that the existing annotations are incomplete, especially regarding 3' UTR annotations. As the reviewer points out, this is most noticeable when using 3'-end biased data and we do not see as severe an issue when analyzing data from a library constructed using a random-primed approach. However, our expanded analyses also clearly point out that each annotation is indeed missing several thousand gene models – a point in the original manuscript that the reviewers regarded with some scepticism. This latter issue would affect RNA-seq analyses regardless of the method of library construction. We also include new data showing that nearly 20% of Ensembl-annotated coding genes completely lack a 3' UTR annotation, which can also affect interpretation from random-primed libraries (an example is given in the revised manuscript).

Our new annotation includes all of the missing genes, along with improved 3' UTR coverage. Our expanded analysis of bulk RNA-seq and integration with scRNA-seq analysis in the revised manuscript also highlights the benefit of having consistent gene calls across experiments. Such integration would be less consistent even with bulk RNA-seq from a random-primed library quantified on current annotations, given that scRNA-seq is typically reliant on accurate 3' UTR models. In addition, our analyses show that it is much more than a "few" genes or just "minor changes" – in nearly every case, we see that RefSeq or Ensembl misses hundreds of genes, many of which are canonical lineage markers, or have major functional relevance.

Finally, we also note a significant increase in the number of transcripts per gene in the new annotation. While a more thorough assessment of transcript diversity is beyond the scope of the current work, our anecdotal observations suggest that there are many more splice isoforms, alternative promoters, and alternative 3' UTRs present in our annotation. We provide some quantification of isoform diversity in the revised manuscript, along with discussion of how to better address this aspect of transcript annotation in future work. Thus, our improved annotation in not simply limited to extended 3' UTR models.

2) The authors state that new transcripts and genes are present in their V4.2 annotation that are missing in both Ensembl and RefSeq. This is based on transcript and gene model output from cufflinks. Some of these models have BLAST hits, which the authors interpret as evidence that these are bona fide expressed transcripts. This is incorrect. Reads can mis-map to processed pseudogenes and other gene remnants and thus look like separate expressed genes. It is therefore not surprising that some of those have BLAST hits. Of the 6562 genes with an XLOC name in v4.2.1.gtf, 4714 are single exon genes. This indicates that the vast majority are indeed likely to be processed pseudogenes. If these gene models were to be used to map against, they would mop up reads from coding genes and distort results. Taken together these models have not passed the many filtering and validation steps using additional evidence (cDNA, cross species comparisons etc) that Ensembl and RefSeq employ. To avoid over-interpretation of these unverified gene models they should be provided separately from those that have Ensembl and/or RefSeq identifiers. The text needs to acknowledge the limitation of these gene models. Claims of missing genes should be made much more cautiously unless independent evidence beyond BLAST hits is provided.

We appreciate the reviewers concerns in this regard and we have made efforts to address this issue. In the revised manuscript, we provide a better description and discussion of "XLOC" entries in the V4 annotations. We also generated an annotation (V4.3) that contains a reduced number of these entries, based on several criteria listed in the manuscript (e.g. high confidence blast hit in zebrafish and/or human; and/or multiple exons, etc). We also include a table with BLAST hits (zebrafish and human) and number of exons for the XLOC loci and whether they were maintained in V4.3. We also make note in the revised manuscript that users who may be concerned with issues regarding mapping and quantification can choose to completely remove these from the annotation.

3) The annotation needs to include existing Ensembl and RefSeq identifiers. Gene symbols are notoriously unreliable and not suitable for comparative analyses.

This issue has been addressed and both Ensembl and RefSeq identifiers have been added to the gene entries. We achieved this through a transcript-based match between the annotations using gffcompare, with subsequent assessment of gene symbols. In some cases, we noted discrepancies, particularly in Ensembl, due to incorrect gene assignments. In these cases, use of the Ensembl identifier would actually result in the wrong gene information being matched, although these cases may be rare. Nonetheless, we also provide a comment regarding heuristic classification for each match that provides a confidence metric regarding the accuracy of the associated Ensembl or RefSeq ID. We refer readers to this as it may lead to improper assignment when importing matched data from Ensembl. We have contacted ZFIN concerning these discrepancies and the cases we have noticed so far have been corrected or noted in the corresponding gene entries.

4) The annotation uses one RNA-seq dataset derived from whole embryos and a limited set of developmental stages. It is possible that the modest improvement is partially due to missing tissue- or stage-specific transcripts and their 3' UTRs. The authors should compare the genes detected in a few tissue-specific datasets with V4.2 to demonstrate sufficient overlap or acknowledge the limitation if a substantial number of genes are absent from the used dataset. Alternatively, the authors could run Cufflinks without Ensembl and RefSeq annotation and then compare how many transcripts overlap Ensembl transcripts. Likewise, alternative polyadenylation and differential 3' end use will affect interpretation of gene expression. This should be discussed.

As described in better detail in the revised manuscript, our approach utilized both an available RNA-seq dataset *and* integration with existing RefSeq and Ensembl annotations. In the revised manuscript we now provide a better description for comparing the resulting annotations and how they address apparent gaps in RefSeq and Ensembl. From these analyses, we show that the V4.3 annotation includes all instances of genes missing from either RefSeq or Ensembl, including any that may have been left out of V4.2. Therefore, there should be no known zebrafish gene annotations missing from V4.3. It is possible that there are additional novel isoforms and 3' UTRs that are present at later or adult stages and these may be missed. This is now mentioned in the Discussion in the revised manuscript.

Technically, the comparison of "genes detected in a few tissue-specific datasets with V4.2 to demonstrate sufficient overlap" is essentially what the scRNA-seq experiment demonstrates. It is clear from those analyses on two distinct cell types (cartilage and epidermis) that V4 can capture all of the changes from Ens95 in addition to 20 to 30% more cluster-enriched genes. In the revised manuscript, we also provide additional data supporting increased gene calls across most scRNA-seq clusters when comparing Ens95 and V4.2 in this dataset. We also now provide bulk RNA-seq analysis of non-endothelial cells using a different transgenic line, as well as improved and expanded analysis of overlap between bulk RNA-seq datasets. These all show similar result across multiple diverse cell types (i.e. that V4 can provide more comprehensive detection of genes compared to Ens95 and RefSeq).

We agree with the reviewers that "alternative polyadenylation and differential 3' end use will affect interpretation of gene expression". If anything, transcriptome annotations with more accurate 3' UTR models, which we believe our annotation captures, will allow more accurate assessment of 3' usage. We would note that accurately capturing alternative polyadenylation and differential 3' UTR usage is highly dependent on stringent technical approaches that reliably capture 3' ends. In any case, given the complexity of this issue, and the increased size of the manuscript due to the need to address other pertinent issues, we have chosen not to extensively discuss this issue, though it is touched upon in the revised Discussion section.

5) The authors need to provide statistics and numbers in the text instead of qualifiers such as "much lower". For example, the authors state "…the overall number of differentially-expressed transcripts identified when using Ensembl was much lower than that found with RefSeq." The difference is 7%. Another example is "increase in the number of median genes". How many? (It's 842 vs. 761.) Likewise, the authors need to provide statistics of their annotation. How many genes are detected, what is the length/ exon number distribution of known transcripts vs. XLOC transcripts etc.

We have made an effort to provide more quantitative descriptors for this and other analyses in the revised manuscript. Although we would note that numbers and statistical analyses are clearly labeled in the figures and all source data is made available.

We also now include better metrics regarding gene, transcript and exon numbers for the V4 annotations, including more definitive comparisons with RefSeq and Ensembl, as well as other descriptors of Ens95 and RefSeq annotations (such as CDS, UTR, etc). These are now included in several new tables in the revised manuscript.

6) A lot of the analysis methodology is missing and no code is provided. "In-house Perl scripts" need to be made available. What were the criteria for "hand-curation"? The same is true for other methods. For example, the intersection analysis of RefSeq and Ens95 is not clear. What does intersection by gene coordinate mean? Is it exactly matching start/end positions or by interval overlap? The discrepancy of 3000 genes is improbably large.

Regarding comparison between transcriptome annotations, we have revised this section considerably. We now provide a more straightforward analysis using gffcompare, a published approach, to perform transcript-level matching. Notably, this more definitive analysis yields a similar number of missing genes as our initial approach.

Therefore, our initial assessment was accurate – that each annotation is missing between 2000 to 3000 genes. We now also include perl and python scripts for custom steps in transcript assembly and an improved description of all computational approaches.

Other required changes and clarifications:1) In Figure 5A, there are 7 cartilage and 9 epidermis genes unique to Ens95 vs. V4.2. Could the authors comment on this? Are these genes bona fide markers of these cell types or false positives? Will the missing genes in V4.2 be incorporated?

In the revised manuscript, these numbers have shifted slightly since we now utilize common gene identifiers rather than gene symbols. We have also provided potential reasons for the identification of genes specifically by Ens95. These are included in a comments column in Figure 5—source data 1, 2. There are a number of reasons these genes show up only in Ens95, including that they fall just on the cusp of statistical cut-offs or are split into 2 immediately adjacent genes in Ens95, but are a single gene (which is also detected in this cluster) in V4.2.

2) Figure 1A, B should be -log.

The incorrect labeling in this case arose from an issue with the DEbrowser package used for analysis. We have now updated the DEbrowser code and corrected the axis labeling throughout the revised manuscript.

3) In Figure 2—figure supplement 1, the 6743 genes from RefSeq only in panel A does not correspond to the 6119 genes (2514+3605) analyzed in panel B. Likewise, the 2514 genes from panel B do not correspond to the 2549 genes (2035+514) in panel C. Please correct or explain these discrepancies in gene number.

In the revised manuscript, we have replaced this analysis with a more definitive and straightforward transcript-matching approach using gffcompare.

4) "Comparison of 3' UTRs for matching genes from the two annotations in those latter cases revealed that the overall differences in length were significant (Figure 2C)." Why has a statistical test been performed here? The authors are not randomly sampling from an underlying population of gene models and then measuring the UTR length. They have selected for the ones that are either longer or shorter in Ensembl 95.

This section has been changed significantly in the revised manuscript. Similar data are now presented in Figure 2—figure supplement 1. The purpose of these graphs is to illustrate the magnitude of difference between 3' UTR annotations in cases where genes from one or another annotation has a longer 3' UTR. The reviewer is correct in that these data point have already been heuristically classified as "longer" or "shorter" obviating the need for a statistical test for comparison. We have made a note to this effect in the corresponding figure legend and have removed the statistical analyses.

5) Figure 3B: If the Ensembl annotation was merged into V4.2, how can V4.2 have shorter UTRs than Ensembl?

These are likely cases where there was not sufficient read depth to support inclusion of a longer 3' UTR model.

6) The counting against the two annotations seems oddly inconsistent. For slc2a1a (Figure 1E and Figure 5D) the same exon portion gets 16 reads in Ens95, but only 4 reads in RefSeq.

The read count in this case is accurate. The actual read depth for *slc2a1a* when including the 3' UTR on V4.3 is >1000. Thus, the level of detection by Ens95 and RefSeq is essentially at the level of noise.

7) It is not clear what is meant by this sentence in the legend for Figure 2—figure supplement 1: "Total numbers of unique genes in each case are not equivalent due to cases there ("where" I assume?) multiple genes from either or both annotations may intersect within a single interval." How many genes does the overlap method fail to distinguish because of this?

As noted above, this analysis has been replaced with an improved method of comparison.

8) "…all of these genes appear preferentially expressed in the erythroid-2 cluster (Figure 7A)". slc25a37 and tfr1a are expressed in more cells at higher levels in the erythroid cluster, not erythroid-2.

Cells positive for these markers are apparent in both clusters, though only a few cells appear to express in the erythroid cluster, while nearly all cells express in erythroid-2.

9) "…mapped onto the RefSeq annotation". Mapping is to the assembly not the annotation, so it should either say "mapped to GRCz11 and counted against the RefSeq annotation" or just "counted against the RefSeq annotation". Same at: "to genome-mapped or ENSEMBL-mapped reads". The reads aren't mapped to the Ensembl annotation, they are counted against it. This needs changing throughout.

This has been corrected throughout the revised manuscript.

10) Figure 3 legend. What do the bars represent in Figure 3C, E and F, median or mean?

For all figures we include a description of error bars and legends.

11) Legends for Figures 4-7. What do the colour bars in the t-SNE expression plots represent? It will be some kind of count, but what exactly?

The colors represent log-transformed normalized expression values. This description has been included in the revised manuscript.

12) It would be interesting to see what the RefSeq and Ensembl comparison looks like for other model organisms (not humans, which has MANE https://www.ncbi.nlm.nih.gov/refseq/MANE/).

We agree that there may be discrepancies between annotations in other model organisms, although such studies are currently beyond the scope of the current work.

13) Figure 5C: There are a lot fewer dots than the Venn diagram would suggest. Are they plotted on top of each other? Jitter might help. Also, what do the genes from the Ens95 counting that don't overlap with the V4.2 ones look like?

For the measurement of UTR lengths, we only consider those genes that are included in the reference geneset, which is where the 3' UTR lengths are derived. Hence, the number is somewhat reduced from the total shown in the Venn. This was originally mentioned in the figure legend. The number of data points is higher in the revised manuscript due to the increased number of reference genes.

14) https://www.umassmed.edu/lawson-lab/transcriptome gives a 404 error.

We apologize for this issue. We generated a simpler URL that redirects to the correct site. The new URL is zf-transcriptome.umassmed.edu

15) Ensembl not ENSEMBL.

This has been corrected.

16) Median ration => Median ratio.

This has been corrected.

17) "In this process, we also manually corrected several incorrect gene names assigned by ENSEMBL that resulted from transcripts overlapping two separate genes, causing a single gene ID to be assigned to two genes." Have these been reported to Ensembl?

We have contacted the Genome Reference Consortium and ZFIN regarding mis-annotations. We have been informed that the discrepancies have been corrected or noted on the corresponding ZFIN pages.

18) tRNAscan is mentioned in the methods, but nowhere else.

Mention of tRNAscan is not relevant and has been removed from the revised manuscript.

19) It would be good if the GTF file contained CDS entries.

We agree on this point and this was under consideration. However, it is not trivial with a large number of potentially new transcripts and new isoforms. Therefore, we have chosen to leave this to future improvements. Inclusion of the matching Ensembl and/or RefSeq IDs in the revised manuscript now allows users to access matching coding sequence information as needed.

20) Figure 2: Venn diagram is incomplete, lacks the non-concordant population

The revised version of the manuscript no longer includes this panel.

21) Figure 2—figure supplement 1 (btw this seems a rather cumbersome label for supplemental figures): the Venn diagrams should be size proportionate.

All Venn diagrams in the manuscript are now size-proportional.

22) Figure 3 and text: reorder bud, shield and dome to dome, shield and bud so as to fully maintain sequence of stages

This has been corrected.

23) Figure 2B: caption should be V4.2

The revised version of the manuscript no longer includes this panel.

24) Figure 2C: scales of charts should be fixed to 15k for direct comparison and to demonstrate length difference of 3'UTR distribution between annotations

Similar panels can now be found in Figure 2—figure supplement 1. All graphs in this figure have the same scale.

25) "…only genes with matched gene symbols between ENSEMBL (v95) and RefSeq were maintained" and "only V4.2 genes that matched by gene symbol were incorporated into the reference set". Why do genes which have gene symbol discrepancies between Ensembl and RefSeq were excluded from the reference set? If the genome location and sequence are the same, this may be a common naming issue, rather than a reason to suggest difference in gene/transcript.

In the revised manuscript, we have changed the reference gene set to gain more comprehensive coverage and provide more accurate comparisons between annotations. We refer reviewers to the Materials and methods section in the revised manuscript for information on how this new gene set was established.

26) The new annotation is not currently accessible at the website cited. Will the authors make an effort to add an improved annotation to Ensembl?

Both the V4.2 and V4.3 annotations, along with associated information tables with RefSeq, Ensembl and ZFIN identifiers, as well as BLAST information for XLOC loci, gene symbols, and descriptions, are available at: zf-transcriptome.umassmed.edu

We will contact both GRC and the UCSC Genome Browser team to request that the annotations be added to the respective browsers. In the past, we have been able to have published datasets added to UCSC, so we expect that this will be possible.

27) Figure 6: add consistent labelling of cell types and gene symbols throughout.

We have addressed this issue in the revised manuscript.